# SPARSE-GUARD: SPARSE CODING-BASED DEFENSE AGAINST MODEL INVERSION ATTACKS

## ABSTRACT

In this paper, we study neural network architectures that are robust to model inversion attacks. It is well-known that standard network architectures are vulnerable to model inversion, where an adversary can reconstruct images or data used to train the network by inspecting the network's output or the intermediate outputs from a single hidden network layer. Surprisingly, very little is known about how a network's architecture contributes to its robustness (or vulnerability). Instead, recent work on mitigating such attacks has focused on injecting random noise into the network layers or augmenting the training dataset with synthetic data.

Our main result is a novel sparse coding-based network architecture, SPARSE-GUARD, that is robust to model inversion attacks. Three decades of computer science research has studied sparse coding in the context of image denoising, object recognition, and adversarial misclassification settings, but to the best of our knowledge, its connection to state-of-the-art privacy vulnerabilities remains unstudied. However, sparse coding architectures suggest an advantageous means to prevent privacy attacks because they allow us to control the amount of irrelevant private information encoded in a model's intermediate representations in a manner that can be computed efficiently during training, that adds little to the trained model's overall parameter complexity, and that is known to have little effect on classification accuracy. Specifically, we demonstrate that compared to networks trained with state-of-the-art noise-based or data augmentation-based defenses, SPARSE-GUARD networks maintain comparable or higher classification accuracy while degrading state-of-the-art training data reconstructions by a factor of 1.2 to 16.2 across a variety of reconstruction quality metrics (PSNR, SSIM, FID) on standard datasets. We also show that SPARSE-GUARD is equally robust to attacks regardless of whether the leaked layer is earlier or later, suggesting it is also an effective defense under novel security paradigms such as Federated Learning.

## 1 INTRODUCTION

The popularization of machine learning has been accompanied by the widespread use of neural networks that were trained on private, sensitive, and proprietary datasets. This has given rise to a new generation of privacy attacks that seek to infer private information about the training dataset simply by inspecting the representation of the training data that remains encoded in the model's parameters (Fredrikson et al., 2015; Gong & Liu, 2016; Kariyappa et al., 2021; Zhong et al., 2022; Mehnaz et al., 2022; Wang et al., 2022; Yuan et al., 2022; Hu et al., 2022; Zhang et al., 2023; Sanyal et al., 2022; Struppek et al., 2022; Carlini et al., 2023; Li et al., 2023).

Of particular concern is a devastating stream of privacy attacks known as model inversion. Model inversion attacks leverage the network's parameters or classifications in order to reconstruct entire images or data that were used to train the network. Early work on model inversion focused on a white-box setting where the attacker has unfettered access to the model or auxiliary information about the training data (Fredrikson et al., 2015; Hitaj et al., 2017; Wang et al., 2019; Zhang et al., 2020; Wei et al., 2020). However, recent work has shown that standard network architectures are vulnerable to model inversion attacks even in the black-box setting where attackers have no knowledge of the model's architecture or parameters, and only have access to the model's classifications

or its intermediate outputs such as leaked outputs from a single hidden network layer (Yang et al., 2019; Mehnaz et al., 2022; Salem et al., 2020; Melis et al., 2019; An et al., 2022; Gong et al., 2023).

Such attacks are feasible because each hidden layer of a standard network architecture captures a detailed representation of the training data. It is well-known that standard dense layers exhibit a tendency to memorize their inputs (Haim et al., 2022; Carlini et al., 2022; Rigaki & Garcia, 2020), so even a minimal leak of intermediate outputs from a single layer are often sufficient to train an inverse mapping for data reconstruction. More concretely, state-of-the-art inversion attacks work by e.g. submitting externally obtained images to the model, observing leaked intermediate layer outputs, then using this data to train a new 'inverted' neural network that reconstructs (predicts) an input image given a leaked output. Such attacks on standard network architectures can reconstruct private training images that are clearly recognizable by humans familiar with the training data (Hitaj et al., 2017; Yang et al., 2019; He et al., 2019; Wei et al., 2020; Aïvodji et al., 2019; Kahla et al., 2022; Struppek et al., 2022; Gong et al., 2023).

Recently, state-of-the-art defenses against model inversion have focused on improving the robustness of standard network architectures by augmenting their training dataset with synthetic data or injecting random noise into the network layers. Specifically, the state-of-the-art defense in Gong et al. (2023) augments the training dataset with GAN-generated fake samples designed to inject spurious features into the trained network that mislead the gradients that are computed during inversion attacks. Alternatively, noise injection-based defenses perturb the network weights or outputs to obfuscate their representations of the training data (Titcombe et al., 2021; Abuadbba et al., 2020; Mireshghallah et al., 2020). Both approaches are costly: data augmentation-based defenses entail the significant computational burden of building a GAN and applying sophisticated parameter tuning techniques during training, and noise-based defenses are known to impose significant reductions in model classification accuracy. Notwithstanding our intuitions from additive noise in other machine learning settings, Differential Privacy guarantees are also known to be inapplicable to protecting the training data representations encoded in a network's layers from model inversion (Wang et al., 2021b; Fredrikson et al., 2014).

*Are different network architectures robust to model inversion attacks?*

Very little is known about how a network's architecture contributes to its robustness (or vulnerability). This is surprising because throughout three decades of research in other application domains such as image denoising (Barlow, 1961; Field, 1994; Chen et al., 2001; Olshausen & Field, 2004; Candès & Donoho, 2004; Rozell et al., 2008; Krause & Cevher, 2010; Ahmad & Scheinkman, 2019), object recognition (Olshausen et al., 1995; Schneiderman, 2004; Kavukcuoglu et al., 2010; Hannan et al., 2023), and adversarial misclassification (Sun et al., 2019; Paiton et al., 2020; Kim et al., 2020; Teti et al., 2022), researchers seeking to control their model's representations of the data have heavily studied sparse coding-based architectures that prune unnecessary features and preserve only the information that is essential to the model objective. Specifically, sparse coding seeks to approximately represent an image (or layer) with only a small set of basis vectors selected from an overcomplete dictionary (Field, 1994; Olshausen & Field, 2004; Candès & Donoho, 2004). While it is well-known that computing a sparse representation using a standard objective function is NP-hard in general (Natarajan, 1995; Davis et al., 1997; Jiang et al., 2012), we now benefit from fast approximation algorithms that generate high-quality sparse representations with little computational overhead (Lee et al., 2006; Rozell et al., 2008; Kavukcuoglu et al., 2010; Krause & Cevher, 2010; Jiang et al., 2012; Mirzasoleiman et al., 2015; Breuer et al., 2020; Chen et al., 2021). Sparse coding architectures leverage this technique by inserting a sparse network layer after a dense layer, such that the sparse layer reduces the dense layer's outputs to a sparse representation.

To our knowledge, sparse coding architectures have not been studied in the context of model inversion or privacy attacks. However, they suggest an advantageous means to prevent such attacks because they control the amount of irrelevant private information encoded in a model's intermediate representations in a manner that can be computed efficiently during training, that adds little to the trained model's overall parameter complexity, and that is known to have little effect on its accuracy.

**Main contribution.** We begin by showing that an off-the-shelf sparse coding-based architecture offers performance advantages compared to state-of-the-art data augmentation and noise-injection based defenses in terms of robustness to model inversion attacks. We then refine this idea to

achieve superior performance. Our main result is a novel sparse-coding based architecture, SPARSE-GUARD, that is robust to state-of-the-art model inversion attacks.

SPARSE-GUARD is defined by pairs of alternating sparse coded and dense layers that jettison unnecessary private information in the input image and ensure that downstream layers do not e.g. reconstruct this information. We show that compared to networks trained with state-of-the-art noise-based or data augmentation-based defenses, SPARSE-GUARD networks maintain comparable or higher classification accuracy while degrading state-of-the-art black-box training data reconstructions by a factor of 1.2 to 16.2 across a variety of reconstruction quality metrics (PSNR, SSIM, FID) on standard evaluation datasets. We emphasize that unlike recent state-of-the-art defenses that require sophisticated parameter tuning techniques to obtain high performance, SPARSE-GUARD obtains these results absent parameter tuning (i.e. using default sparsity parameters) due to the natural robustness properties of sparse coded layers. We also show that SPARSE-GUARD is equally robust regardless of whether the leaked layer is an earlier layer or a later layer. This consistency is desirable both because model inversion attacks are known to work better on earlier hidden layers due to their greater similarity to training data (He et al., 2019), and because it suggests that SPARSE-GUARD is also an effective defense for novel security paradigms such as Federated Learning, as we discuss below.

More broadly, our results show a deep connection between state-of-the-art machine learning privacy vulnerabilities and three decades of computer science research on sparse coding for other application domains. We provide a cluster-ready PyTorch codebase to encourage further research in this regard.

**Paper organization.** Section 2 describes our adversarial settings. Section 3 describes the SPARSE-GUARD architecture and its associated sparse coding technique. Section 4 compares the performance of SPARSE-GUARD and its variants to state-of-the-art alternatives on standard evaluation datasets in end-to-end and split-network settings. Section 5 provides an empirical analysis of why sparse coding prevents model inversion attacks. Section 6 concludes.

## 2 ADVERSARIAL SETTINGS: BLACK-BOX SPLIT & END-TO-END ATTACKS

We consider settings that capture 'worst-case' black-box attacks with a powerful attacker. Specifically, our setting is black-box because we suppose that attackers have no knowledge of model architecture or parameters. However, we suppose the attacker has access to raw, high-dimensional intermediate outputs such as leaked outputs from a single hidden network layer. This setting captures the 'worst-case' where the attacker has direct access to the area of the target model that stores private information about the training data. In other realistic settings, black-box attackers may instead observe only (low-dimensional) model classifications. However, a good defense in our setting reflects robustness to even strong black-box attacks in the presence of leaks. We consider two variants:

**End-to-end network setting.** Our primary setting is the standard end-to-end network setting where the attacker accesses the *last* hidden layer's outputs (Wang & Wang, 2022; Song & Mittal, 2021).

**Split network setting (Federated Learning).** We also consider the split network setting described by Titcombe et al. (2021) where the attacker has access to raw intermediate outputs from an *earlier* layer. This setting is relevant for two reasons. First, there has been much recent interest in Federated Learning (collaborative learning) architectures that split the network across multiple agents (Konečný et al., 2016; McMahan et al., 2017; Bonawitz et al., 2019). Such architectures can enable learning in privacy-fraught domains such as medicine where legal requirements limit data sharing (Vepakomma et al., 2018; Kaissis et al., 2020). However, it is now well-known that Federated Learning architectures (and split networks in particular) are susceptible to model inversion attacks (Titcombe et al., 2021). Defenses for such learning settings are urgently needed.

Second, model inversion attacks are known to be more effective when the attacker has access to outputs from earlier layers, as earlier layers may exhibit a more direct representation of the input images (He et al., 2019). To address the 'worst-case' of this vulnerability, we consider the setting where the attacker has access to raw intermediate outputs from the *first* linear network layer.

## 3 THE SPARSE-GUARD ARCHITECTURE.

We now describe the SPARSE-GUARD architecture, which is defined by alternating pairs of Sparse Coding Layers (SCL) and dense layers, followed by downstream linear and/or convolutional layers.

**Sparse Coding Layer (SCL).** Sparse coding converts raw inputs to sparse representations, i.e. representations where only a few neurons whose features are useful in reconstructing the inputs are active. Our Sparse Coding Layer (SCL) performs sparse coding to obtain a sparse representation of a previous dense layer's representation (if the SCL is not the first layer in the network) or of the inputs (if the SCL is the first layer in the network). We illustrate the working principle of SCL in Fig. 2.

Formally, each SCL performs a reconstruction minimization problem to compute the sparse representation of its inputs (either a previous layer's representation or of the inputs to the network). Suppose the input to a (2D convolutional) SCL is $\mathcal{X} \in \mathbb{R}^{\mathcal{C} \times \mathcal{H} \times \mathcal{W}}$ with $\mathcal{H}$ height, $\mathcal{W}$ width, and $\mathcal{C}$ channels/features. The goal is

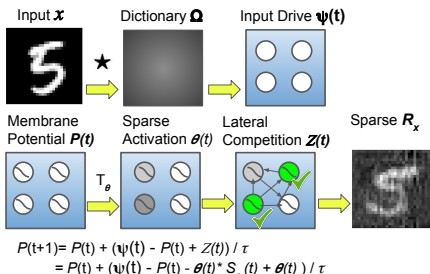

Figure 1: Pipeline of neuron (membrane potential) dynamics in Sparse Coding Layer (SCL) with lateral competitions.

to find the sparse representation $\mathcal{R}_x \in \mathbb{R}^{\mathcal{F} \times \lfloor \mathcal{H}/S_h \rfloor \times \lfloor \mathcal{W}/S_w \rfloor}$, where $\mathcal{R}_x$ has few active neurons and corresponds to a denoised version of the input $\mathcal{X}$, and $S_w$ and $S_h$ indicate convolutional strides across the width and height of the input, respectively. $F$ is the number of convolutional features in the SCL layer's dictionary, $\Omega \in \mathbb{R}^{\mathcal{F} \times \mathcal{C} \times \mathcal{H}_f \times \mathcal{W}_f}$, where $\mathcal{H}_f$ and $\mathcal{W}_f$ are the height and width of each convolutional feature, respectively. Per Figure 1, the sparse coding layer starts with its input, $\mathcal{X}$, and dictionary of features, $\Omega$, to produce $\mathcal{R}_x$ by solving the following sparse reconstruction problem:

$$\min_{\mathcal{R}_x} \frac{1}{2} ||\mathcal{X} - \mathcal{R}_x \circledast \Omega||_2^2 + \lambda ||\mathcal{R}_x||_1 \tag{1}$$

where the first term represents how much information is preserved about $\mathcal{X}$ by $\mathcal{R}_x$ by measuring the difference between $\mathcal{X}$ and its reconstruction, $\mathcal{R}_x \circledast \Omega$, computed with a transpose convolution, $\circledast$. The second term measures how sparse $\mathcal{R}_x$ is, and $\lambda$ is a constant which determines the tradeoff between reconstruction fidelity and sparsity. Equation 1 is convex in $\mathcal{R}_x$, meaning we will always find the optimal $\mathcal{R}_x$ that solves Equation 1.

Among different techniques to perform sparse coding, we leverage the commonly used Locally Competitive Algorithm (LCA) (Rozell et al., 2008). LCA implements a recurrent network of leaky integrate-and-fire neurons that incorporates the general principles of thresholding and feature-similarity-based competition between neurons to solve Equa-

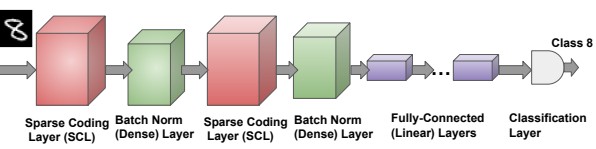

Figure 2: Architecture of SPARSE-GUARD.

tion 1. Although Rozell introduced LCA in the non-convolutional setting, it can (and has been Teti et al. (2022); Kim et al. (2020)) readily adapted to the convolutional setting (see Section A.1 for details). Specifically, each LCA neuron has an internal membrane potential $\mathcal{P}$ which evolves per the following differential equation:

$$\dot{\mathcal{P}}(t) = \frac{1}{\tau} [\Psi(t) - \mathcal{P}(t) - \mathcal{R}_x(t) * \mathcal{G}] \tag{2}$$

where $\tau$ is a time constant, $\Psi(t) = \mathcal{X} * \Omega$ is the neuron's bottom-up drive from the input computed by taking the convolution, $*$, between the input, $\mathcal{X}$, and the dictionary, $\Omega$, and $-\mathcal{P}(t)$ is the leak term. Lateral competition between neurons is performed via the term $-\mathcal{R}_x(t) * \mathcal{G}$, where $\mathcal{G} = \Omega * \Omega - I$ is the similarity between each feature and the other $\mathcal{F}$ features ($-I$ prevents self interactions). $\mathcal{R}_x$ is computed by applying soft threshold activation $T_\lambda(x) = \text{relu}(x - \lambda)$ to the neuron's membrane potential, which produces nonnegative, sparse representations. Overall, this means that in LCA neurons will compete to determine which ones best represent the input and, thus, will have non-zero activations in $\mathcal{R}_x$, the output of the SCL that is passed to the next layer.

**SPARSE-GUARD architecture.** The SPARSE-GUARD architecture is defined by the use of multiple *pairs of sparse coding and dense (batch norm) layers* after the input image, which can then be followed by other (linear, convolutional) layers. Fig. 2 illustrates this design principle. The *key intuition* is that the first sparse layer jettisons unnecessary private information in the input image. Then,

by alternating sparse-dense pairs of layers, we ensure that unnecessary information is also jettisoned from downstream layers. In this manner, downstream layers also do not convey unnecessary private information to the adversary, and they also do not e.g. learn to reconstruct private information jettisoned by the first sparse layer. In short, previous defenses work by trying to mislead attackers by pushing model features in a wrong direction, either randomly via noise or strategically via adversarial examples. In contrast, SPARSE-GUARD removes the unnecessary private information.

Training SPARSE-GUARD is identical to training a standard network with one exception. Specifically, after each backpropagation updates non-sparse layers, we perform a fast update on the sparse layers, except for the very first sparse layer that sparse-codes the image input.[1]

**SPARSE-GUARD training complexity & large scale applications.** While we focus on the neuron lateral competition approach to sparse coding as it is practically convenient and well-represented in recent work (Teti et al., 2022), we note that for large-scale machine learning applications, we now have practical parallel algorithms that learn the sparse coding dictionary near-optimally w.p. in parallel time (adaptivity) that is logarithmic in the size of the data (Jiang et al., 2012; Breuer et al., 2020; Chen et al., 2021). Fast single-iteration heuristics are also available (see e.g. Wu et al. (2020)). Thus, even for large-scale applications, computing sparse representations while training SPARSE-GUARD adds little computational overhead compared to sophisticated optimization-based techniques necessary for recent defenses (Gong et al., 2023). In practice, even our basic sparse coding research implementations (see Section 4 and Appendix A.2 below) are slightly faster than highly optimized Torch implementations of GAN-based defenses.

## 4 EXPERIMENTS

Our goal in this section is to show that SPARSE-GUARD performs well compared to state-of-the-art data augmentation and noise-based defenses as well as practical defenses commonly used in leading industry models in terms of both classification accuracy and a variety of attack reconstruction quality metrics. To accomplish this, we conduct two sets of experiments. In the first set, we compare SPARSE-GUARD networks to a variety of baselines in terms of their robustness to a state-of-the-art attack that leverages leaked outputs from the networks' last hidden layer. This allows us to assess SPARSE-GUARD's defenses in a realistic black-box end-to-end network setting.

In the second set of experiments, test SPARSE-GUARD and baselines in a split network setting where the attacker has black-box access to leaked outputs from the *first* linear network layer. Robustness in this setting is desirable both because model inversion attacks are known to be more effective on earlier hidden layers (He et al., 2019), and also because an algorithm that is robust to such attacks would be an effective defense under novel security paradigms such as Federated Learning, which is known to be vulnerable to model inversion attacks (Titcombe et al., 2021).

In all experiments, we consider the simplest case of SPARSE-GUARD architecture that contains SPARSE-GUARD's alternating sparse-and-dense layer pairs followed by only linear layers. We note that adding downstream convolutional layers or more sophisticated downstream architectures is certainly possible, though we avoid this here in order to compare the essence of the SPARSE-GUARD approach to the benchmarks. Appendix A.3 describes SPARSE-GUARD architecture details. In the split network setting, we are careful to use slightly shallower SPARSE-GUARD architectures with fewer linear layers to match the split network experiments of Titcombe et al. (2021).

**SPARSE-GUARD *without* parameter tuning.** Recent state-of-the-art defenses such as GAN-based defenses require sophisticated automatic parameter tuning techniques such as focal tuning and continual learning to obtain high performance (Gong et al., 2023). To test whether SPARSE-GUARD can be effective *absent* parameter tuning, we just run SPARSE-GUARD with sparsity parameter $\lambda$ set to $0.1$, $0.25$, or $0.5$—the default values from various sparse coding contexts.

**Defense baselines.** We compare SPARSE-GUARD to six baselines, including state-of-the-art defenses and practical defenses commonly deployed in leading industry models:

---

[1]We can optionally also allow backpropagation to update this very first sparse layer after the input image. We do this in our experiments. Alternatively, in some learning scenarios it may be advantageous to instead precompute the sparse representation of each image and delete the original images before training, as the first sparse layers remain fixed when we optionally exclude them from backpropagation.

- **Laplace-Noise** (Titcombe et al., 2021). We train a state-of-the-art Laplace $\mathcal{L}(\mu{=}0, b{=}0.5)$ noise defense as in Titcombe et al. (2021). We also try more noise—see Appendix 6 and 7
- **GAN-Opt** (Gong et al., 2023). We train the state-of-the-art defense from Gong et al. (2023) that uses sophisticated tuning and two types of GAN-generated images. We also compare against a '**++**' version that adds extra Continual Learning accuracy optimizations.
- **Sparse-Standard**. We train an off-the-shelf sparse coding architecture (Teti et al., 2022) with one sparse layer after the input image via lateral competition as in SPARSE-GUARD.
- **GAN**. We train a GAN for 25 epochs to generate fake samples, then train the target model with both original and GAN-generated samples. This defense is frequently used in industry.
- **Gaussian-Noise**. We draw random noises from the normal distribution $\mathcal{N}(\mu{=}0, \sigma{=}0.5)$ and inject them into an intermediate dense layer after training (a common defense in industry).
- **No-Defense**. The baseline target model with no added defenses.

**Performance metrics.** We measure the quality of the attacker's training data reconstructions using a variety of standard metrics. Let $X_{in}^*$ denote the reconstruction of training image $X_{in}$. Then:

- **Peak signal-to-noise ratio (PSNR)** [*lower = better*].
  PSNR captures the ratio of maximum squared pixel fluctuations between $X_{in}$ and $X*_{in}$ over mean squared error (MSE).
- **Structural similarity (SSIM)** (Wang et al., 2004) [*lower = better*].
  $SSIM(X_{in}, X_{in}^*) = l_{dis}(X_{in}, X_{in}^*)c_{dis}(X_{in}, X_{in}^*)c_{loss}(X_{in}, X_{in}^*)$. SSIM measures distortion in $X_{in}^*$ as a product of luminance distortion, contrast distortion, & correlation loss.
- **Fréchet inception distance (FID)** (Heusel et al., 2017) [*higher = better*].
  $FID(X_{in}, X_{in}^*) = ||\mu_{X_{in}} - \mu_{X_{in}^*}||^2 + Tr(\text{Cov}_{X_{in}} + \text{Cov}_{X_{in}^*} - 2*\sqrt{\text{Cov}_{X_{in}} \cdot \text{Cov}_{X_{in}^*}})$
  FID measures reconstruction quality as a distributional difference between $X_{in}$ and $X_{in}^*$.

**Attack.** We consider a state-of-the-art surrogate model training attack optimized via SGD (Xu et al., 2023; Aïvodji et al., 2019). This attack works by querying the target model with an externally obtained dataset (in this case, a holdout set from the experiment dataset). The attack then uses the corresponding model outputs to train an inverted surrogate model that outputs actual training data. We provide attack details in the Appendix A.4.

**Target model.** We focus on privacy attacks on linear networks because they capture the essence of the privacy attack vulnerability (Fredrikson et al., 2015; Hidano et al., 2017), and because there is broad consensus that a principled understanding of their emerging privacy (and security) vulnerabilities[2] is urgently needed(Sannai, 2018; Liu et al., 2019; Wu et al., 2022; Heredia et al., 2023).

**Datasets.** We test our performance on the two standard datasets most commonly used to benchmark model inversion attacks: MNIST and Fashion MNIST (Zhang et al., 2020; Salem et al., 2020; Tian et al., 2022; Aïvodji et al., 2019; Wei et al., 2020; Hitaj et al., 2017; Titcombe et al., 2021; Wang et al., 2019; He et al., 2019; Erdoğan et al., 2022).

**PyTorch codebase and experimental setup.** For the experiments, we consider the standard train test split of 70% and 30%. After training each defense model, we run attacks to reconstruct the entire training set and compare reconstruction performance. We run all the experiments on a standard industry production cluster with 4 nodes and DELL Tesla V100 GPUs with 40 cores. *We provide a full (author-anonymized) PyTorch codebase that implements attacks,* SPARSE-GUARD *and its associated sparse coding architecture, other defenses, and replication codes for our experiments at:* https://anonymous.4open.science/r/sparse-guard-EE8C/.

### 4.1 RESULTS OF EXPERIMENTS SET 1: END-TO-END NETWORKS

Table 1 reports reconstruction quality measures and accuracy for SPARSE-GUARD and benchmarks on both datasets in the end-to-end network setting (*lower rows = better defense performance*). Fig. 3 shows the reconstructions of three images (sampled uniformly at random) under different defenses.

---

[2]We also note that results on linear models may generalize better than results on more application-specific models, and linear models trained on private data remain ubiquitous among top industry products.

Table 1: Experiments set 1: Performance in *end-to-end* network setting *(lower rows=better defense)*.

| Dataset | Defense | PSNR ⇊ | SSIM ⇊ | FID ($10^3$) ⇈ | Accuracy |
|---|---|---|---|---|---|
| MNIST | NO-DEFENSE | 40.87 | 0.982 | 16.31 | 0.971 |
| | GAUSSIAN-NOISE | 40.88 | 0.983 | 15.88 | 0.958 |
| | GAN | 40.69 | 0.981 | 16.59 | 0.968 |
| | Titcombe et al. (2021) | 31.18 | 0.863 | 47.32 | 0.980 |
| | Gong et al. (2023)++ | 30.37 | 0.838 | 72.99 | 0.987 |
| | Gong et al. (2023) | 29.05 | 0.817 | 75.39 | 0.985 |
| | SPARSE-STANDARD | 21.34 | 0.439 | 142.9 | 0.986 |
| | **SPARSE-GUARD0.1** | **19.54** | **0.502** | **178.5** | **0.984** |
| | **SPARSE-GUARD0.25** | **18.81** | **0.340** | **174.1** | **0.983** |
| | **SPARSE-GUARD0.5** | **17.85** | **0.164** | **335.5** | **0.977** |
| Fashion MNIST | NO-DEFENSE | 37.86 | 0.975 | 13.91 | 0.886 |
| | GAUSSIAN-NOISE | 36.54 | 0.969 | 16.49 | 0.815 |
| | GAN | 37.68 | 0.974 | 19.26 | 0.883 |
| | Gong et al. (2023)++ | 27.71 | 0.794 | 41.35 | 0.906 |
| | Titcombe et al. (2021) | 26.66 | 0.759 | 53.76 | 0.905 |
| | Gong et al. (2023) | 21.24 | 0.523 | 93.08 | 0.888 |
| | SPARSE-STANDARD | 19.35 | 0.446 | 128.4 | 0.879 |
| | **SPARSE-GUARD0.1** | **17.92** | **0.209** | **196.1** | **0.897** |
| | **SPARSE-GUARD0.25** | **17.03** | **0.186** | **195.2** | **0.887** |
| | **SPARSE-GUARD0.5** | **14.51** | **0.069** | **423.2** | **0.876** |

Observe that training data reconstructions under the *least sparse* version SPARSE-GUARD0.1 are degraded by a factor of 1.5 to 3.8 vs. LAPLACE-NOISE (Titcombe et al., 2021), and by a factor of 1.2 to 4.7 vs. the two optimized GAN defenses of (Gong et al., 2023) across the quality metrics. Increasing SPARSE-GUARD's sparsity $\lambda$ to 0.5 widens the performance gap, increasing these factors to 1.8 to 11.0 and 1.5 to 11.5, respectively. SPARSE-GUARD and benchmarks all outperform GAN and GAUSSIAN-NOISE defenses common in industry across all metrics (including accuracy).

It is clear that SPARSE-GUARD's large improvements in reconstruction metrics also do not come at the cost of accurate classification. Observe that SPARSE-GUARD0.1's accuracy is better than that of Gong et al. (2023), and worse by a (negligible) factor of 0.0035 compared to the '++' version of Gong et al. (2023) that uses extra continual learning based optimization to improve accuracy (we do not add extra optimization techniques to SPARSE-GUARD, as our goal is to focus specifically on the performance of the sparse coding approach). SPARSE-GUARD0.1's accuracy is also comparable (slightly better on MNIST, slightly worse on FMNIST) to that of Titcombe et al. (2021). In Appendix B, we also try increasing the Titcombe et al. (2021) noise parameter, but this results in a significant accuracy drop without matching SPARSE-GUARD's reconstruction metrics.

**SPARSE-GUARD VS. SPARSE-STANDARD.** Interestingly, our SPARSE-STANDARD baseline outperforms Laplace-based and optimized GAN-based defenses by a factor of 1.38 to 3.02 and 1.1 to 3.11 respectively, though it has worse SSIM and FID compared to SPARSE-GUARD0.5 by factors of 2.34 to 6.46 and 2.34 to 3.29, respectively (and slightly worse PSNR). Recall that SPARSE-STANDARD computes just a single sparse layer after the input image. Thus, each image's sparse representation can be precomputed and SPARSE-STANDARD can then be trained by an off-the-shelf

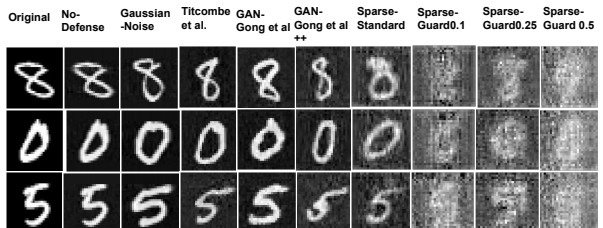

Figure 3: Original images and reconstructed images under SPARSE-GUARD and benchmarks.

Table 2: Experiments set 2: Performance in *split* network setting *(lower rows=better defense)*.

| Dataset | Defense | PSNR ↓↓ | SSIM ↓↓ | FID ($10^3$) ↑↑ | Accuracy |
|---------|---------|---------|---------|-----------------|----------|
| MNIST | No-Defense | 31.21 | 0.923 | 19.64 | 0.963 |
| | Gaussian-Noise | 31.07 | 0.922 | 23.27 | 0.972 |
| | GAN | 28.39 | 0.894 | 27.26 | 0.969 |
| | Gong et al. (2023) | 28.30 | 0.806 | 69.38 | 0.986 |
| | Titcombe et al. (2021) | 25.40 | 0.713 | 76.88 | 0.952 |
| | Gong et al. (2023)++ | 21.94 | 0.591 | 97.33 | 0.991 |
| | Sparse-Standard | 18.71 | 0.288 | 188.4 | 0.981 |
| | **Sparse-Guard0.1** | **16.17** | **0.109** | **227.4** | **0.988** |
| | **Sparse-Guard0.25** | **17.40** | **0.058** | **301.6** | **0.980** |
| | **Sparse-Guard0.5** | **14.98** | **0.044** | **307.7** | **0.975** |
| Fashion MNIST | No-Defense | 29.66 | 0.911 | 14.33 | 0.868 |
| | Gaussian-Noise | 29.49 | 0.909 | 14.81 | 0.871 |
| | GAN | 26.03 | 0.849 | 19.33 | 0.885 |
| | Gong et al. (2023)++ | 25.77 | 0.726 | 57.72 | 0.908 |
| | Gong et al. (2023) | 23.70 | 0.631 | 97.52 | 0.884 |
| | Titcombe et al. (2021) | 20.48 | 0.565 | 81.01 | 0.872 |
| | Sparse-Standard | 19.54 | 0.405 | 200.5 | 0.882 |
| | **Sparse-Guard0.1** | **18.11** | **0.154** | **171.1** | **0.904** |
| | **Sparse-Guard0.25** | **17.74** | **0.188** | **203.8** | **0.896** |
| | **Sparse-Guard0.5** | **17.15** | **0.134** | **270.4** | **0.879** |

optimizer as there are no sparse coding updates. Therefore, while Sparse-Standard offers an inferior defense vs. Sparse-Guard, it nonetheless offers a fast and practical defense for less privacy-critical application domains that do not merit even the modest additional training effort required to update Sparse-Guard's other sparse layers. The fact that a simplistic sparse coding approach already conveys performance advantages over much more sophisticated defenses underscores the natural connection between sparse representations and training data privacy.

**Sparse-Guard's sparsity vs. defense: studying the sparsity parameter $\lambda$.** Table 3 shows that for each $\lambda$ and defense metric, Sparse-Guard significantly outperforms the off-the-shelf Sparse-Standard architecture at the cost of a small decrease in accuracy. As such, for a given $\lambda$ with Sparse-Standard, we can use a (smaller) $\lambda$ with Sparse-Guard to obtain better reconstruction *and* higher or comparable (within 0.0017) accuracy. Sparse-Guard is also amenable to far more sophisticated tuning (and performance improvements) by tuning different $\lambda$ for each sparse layer (for example, by having a sparser representation of the input image but a less sparse reduction of a downstream layer). We avoid such tuning here as it is unnecessary to achieve good performance.

Table 3: Sparse-Standard and Sparse-Guard performance with $\lambda \in \{0.1, 0.25, 0.5, 0.75\}$

| | PSNR↓↓ | | SSIM↓↓ | | FID ($10^3$)↑↑ | | Accuracy | |
|---|--------|--------|--------|--------|--------|--------|--------|--------|
| $\lambda$ | Sp-Std | Sp-Guard | Sp-Std | Sp-Guard | Sp-Std | Sp-Guard | Sp-Std | Sp-Guard |
| 0.1 | 23.45 | 19.54 | 0.650 | 0.502 | 111.5 | 178.5 | 0.984 | 0.984 |
| 0.25 | 21.34 | 18.81 | 0.438 | 0.340 | 142.9 | 174.1 | 0.986 | 0.983 |
| 0.5 | 22.16 | 17.85 | 0.598 | 0.164 | 136.9 | **335.4** | 0.985 | 0.977 |
| 0.75 | 22.39 | **14.65** | 0.593 | **0.086** | 142.0 | 214.1 | 0.981 | 0.971 |

## 4.2 Results of experiments set 2: Split networks

Table 2 reports performance of Sparse-Guard and benchmarks on both datasets in the split-network setting. Here, training data reconstructions under Sparse-Guard0.1 are degraded by a factor of 1.1 to 6.5 compared to the Laplace noise approach of Titcombe et al. (2021) and by a factor of 1.3 to 7.4 compared to GAN defenses of Gong et al. (2023). Sparse-Guard0.5 outperforms the same benchmarks by factors of 1.2 to 16.2 and 1.5 to 18.3, respectively. Importantly, Sparse-Guard's performance in this split-network setting is comparable to the end-to-end network setting. This suggests that Sparse-Guard is also effective under novel security paradigms such as Federated Learning, which may be vulnerable to leaks from earlier layers.

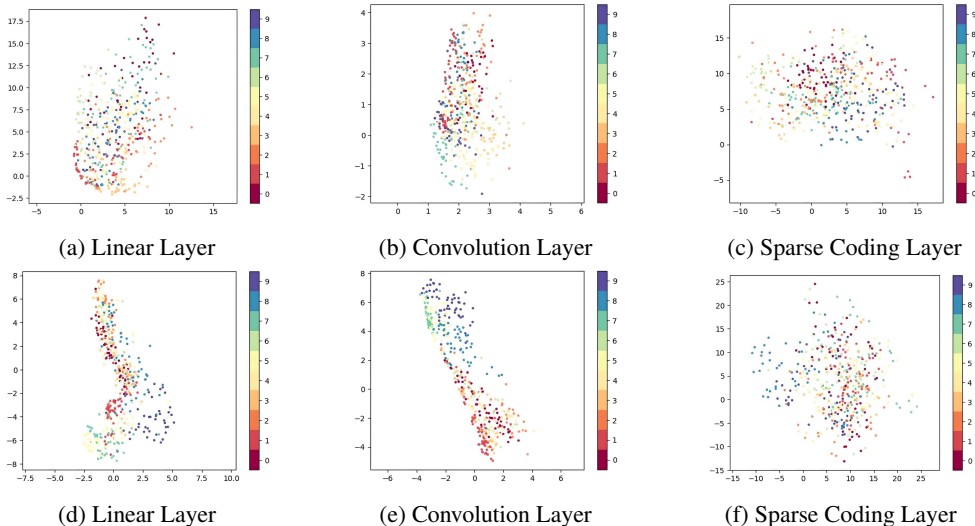

(a) Linear Layer        (b) Convolution Layer        (c) Sparse Coding Layer

(d) Linear Layer        (e) Convolution Layer        (f) Sparse Coding Layer

Figure 4: UMap 2D projections of input images' features by class after 2 linear layers, 2 convolutional layers, or 2 sparse-coded layers. The top row plots MNIST & bottom row is Fashion MNIST.

## 5 Empirical Analysis of Sparse Coding Layer Robustness to Attack

Sparse-coding layers' robustness to privacy attacks can be observed empirically. Consider that in both end-to-end and split network settings, the attacker trains the attack to map leaked raw hidden layer outputs back to input images. Attacks are thus highly dependent on these outputs' distributions. To visualize these distributions, recall that UMAP projections compute a 2D visualization of the global structure of distances between different training images' features according to a particular layer (McInnes et al., 2018). Fig. 4 plots UMAP 2D projections of linear layer feature distributions of each training data input *after* either two linear layers (Figs. 4a & 4d), two convolutional layers (Figs. 4b & 4e), or two sparse coding layers (with interspersed dense layers – Figs. 4c & 4f).

Importantly, observe that after two linear or two convolutional layers, points are clustered by color, meaning that input images' features are highly clustered by label (e.g. in MNIST nearly all 4's have similar features). This class-clustered property leaves such layers vulnerable to model inversion attacks, as an attacker can 'home in on' examples from a specific class. In contrast, the goal in sparse coding is not to optimize the classification objective by separating classes, but rather to jettison unnecessary information. Here, this means that unnecessary information is jettisoned both from the input image and also the downstream dense layer. Per Figs. 4c & 4f, this tends to 'uncluster' remaining non-sparsified features of training examples from the same class, making it significantly harder for an attacker to compute informative gradients used to 'home in on' a training example.

## 6 Conclusion

In this paper, we have provided the first study of neural network architectures that are robust to model inversion attacks. We have shown that a standard off-the-shelf sparse-coding architecture obtains performance similar to state-of-the-art defenses, and we have refined this idea to design an architecture that obtains superior performance. More broadly, we have shown that the natural properties of sparse coded layers can control the extraneous private information about the training data that is encoded in a network without resorting to complex and computationally intensive parameter tuning techniques. Our work reveals a deep connection between state-of-the-art privacy vulnerabilities and three decades of computer science research on sparse coding for other application domains.

## 7    REPRODUCIBILITY

Full cluster-ready PyTorch (Paszke et al., 2019) implementations of SPARSE-GUARD and all benchmarks as well as replication codes for all experiments can be found on our (author-anonymized) repository at: https://anonymous.4open.science/r/sparse-guard-EE8C/.

We provide full details of the cluster hardware and all parameter choices used in our experiments in Appendix A.2 and A.3, and in Appendix Tables 4 and 5.

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

## A APPENDIX

### A.1 ADAPTING ROZELL LCA TO CONVOLUTIONAL NETWORKS

Although the original LCA formulation (Rozell et al., 2008) was introduced for the non-convolutional case, it is based on the general principle of feature-similarity-based competition between neurons within the same layer, which can be (and has been (Teti et al., 2022; Kim et al., 2020)) adapted to the convolutional setting via only two minimal changes to Equation 2. In Rozell's original formulation, $\Psi(t)$ can simply be recast from a matrix multiplication to a convolution between the input and dictionary. Second, the lateral interaction tensor, $\mathcal{G}$ in Equation 2, can also be recast from a matrix multiplication to a convolution between the dictionary and its transpose.

### A.2 CLUSTER DETAILS

We run all our experiments using the slurm batch jobs on industry-standard high-performance GPU clusters with 40 cores and 4 nodes. Details of the hardware and architecture of our cluster are described in Table 4. We note that noise-based defenses are typically fastest on this architecture (though they are the worst-performing), closely followed by SPARSE-STANDARD, then SPARSE-GUARD and (Gong et al., 2023). We emphasize that our sparse coding implementations are 'research-grade', unlike the optimized torch GAN implementations available for (Gong et al., 2023). For large scale applications, SPARSE-GUARD's sparse coding updates can be accelerated such that they can be computed extremely efficiently (see the training complexity discussion at the end of Section 3).

### A.3 PARAMETERS AND ARCHITECTURE OF THE PROPOSED SPARSE-GUARD

We implement SPARSE-GUARD using two Sparse Coding Layers (SCL): One following the input image, and one following a downstream dense batch normalization layer. Finally, we follow these two pairs of dense-then-sparse layers with downstream fully connected (linear) layers before the classification layer. In the case of end-to-end network experiments, we use 5 downstream linear layers, which is a reasonable default. In the split network setting, we are careful to use 3 downstream fully connected layers in order to match the architectures used in the split network experimental setup of (Titcombe et al., 2021), and per our public codebase, we make every effort to make the benchmarks within each setting comparable in terms of architecture, aside from the obvious difference of SPARSE-GUARD's sparse layers We train SPARSE-GUARD's sparse layers with 500 iterations of lateral competitions during reconstructions in SCL layers. We emphasize that SPARSE-GUARD can be made significantly more complex, either via the addition of more sparse-dense pairs of layers, or by adding additional (convolutional, linear) downstream layers before classification. We avoid such complexity in the experiments in order to compare more directly to benchmarks and because our goal is to study an architecture that captures the essence of SPARSE-GUARD. We give all parameter and training details in Table 5.

Table 4: Hardware Details of the Cluster in our Experiments.

| $Parameter$ | MEASUREMENTS |
|---|---|
| Core | 40 |
| RAM | 565GB |
| GPU | Tesla V100 |
| Nodes | p01-p04 |
| Space | 1.5TB |

### A.4 MODEL INVERSION ATTACK METHODOLOGY: ADDITIONAL DISCUSSION

Because privacy attacks are an emerging field, we feel it is relevant to include additional context and discussion here. Recent work has highlighted a variety of attack vectors targeting sensitive training data of machine learning models Liu et al. (2022); Dibbo et al. (2023); Vhaduri et al. (2021); Tramèr et al. (2022); Shokri et al. (2017); Zhang et al. (2020); Choquette-Choo et al. (2021); Dibbo (2023); Vhaduri et al. (2022); Sablayrolles et al. (2019); Gong & Liu (2016); Zhong et al. (2022); Carlini et al. (2023); Vhaduri et al. (2023); Li et al. (2023); Carlini et al. (2021). Adversaries with different access (i.e., black-box, white-box) to these models perform different attacks leveraging a wide range

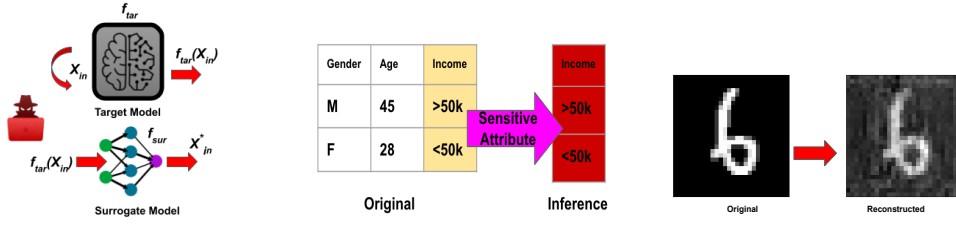

(a) Model Inversion Attack     (b) Attribute Inference     (c) Image Reconstruction

Figure 5: Illustration of Model Inversion attack along with (a.) pipelines–an adversary queries target model $f_{tar}$ with inputs $\mathcal{X}_{in}$ to obtain output $f_{tar}(X_{in})$. Then adversary trains a surrogate attack model $f_{sar}$, where the $f_{tar}(X_{in})$ is the input and $\mathcal{X}^*$ is the output; and (b.) categories, i.e., attribute inference (AttrInf) attack, where the adversary infers sensitive attribute $\mathcal{X}_s$ with or without knowing non-sensitive attribute values, i.e., $\mathcal{X}_{ns} \rightarrow \mathcal{X}_s$ and (c.) image reconstruction (ImRec) attack, where adversary reconstructs similar to original images, i.e., $\mathcal{X}_{in} \approx \mathcal{X}_{in}^*$.

of capabilities, e.g., knowledge about the target model confusion matrix and access to blurred images of that particular class Fredrikson et al. (2015); Choquette-Choo et al. (2021); He et al. (2019); Wang et al. (2021a); Juuti et al. (2019). Such attacks commonly fall under the umbrella of privacy attacks, which include specific attacker goals such as membership inference, model stealing, model inversion, etc. Mehnaz et al. (2022); Wang et al. (2022); Yuan et al. (2022); Hu et al. (2022).

Our focus is model inversion attack, where an adversary aims to infer sensitive training data attributes $X_s$ or reconstruct training samples $X_{in}$, a severe threat to the privacy of training data $D_{Tr}$ Titcombe et al. (2021); Mehnaz et al. (2022). In Figure 5a, we present the pipelines of the model inversion attack. Depending on data types and purpose, model inversion attacks can be divided into two broader categories: (i) attribute inference (AttrInf) and (ii) image reconstruction (ImRec) attacks Dibbo (2023). In AttrInf attacks, it is assumed the adversary can query the target model $f_{tar}$ and design a surrogate model $f_{sur}$ to infer some sensitive attributes $X_s$ in training data $D_{Tr}$, with or without knowing all other non-sensitive attributes training data $X_{ns}$ in the training data $D_{Tr}$, as presented in Figure 5b. In ImRec attacks the adversary reconstructs entire training samples $D_{Tr}$ using the surrogate model $f_{sur}$ with or without having access to additional information

Table 5: Architecture and Parameters of SPARSE-GUARD implementation.

| $Parameter$ | VALUE |
|---|---|
| Sparse Layers | 2 |
| Batch Norm Layers | 2 |
| Fully Connected Layers | 5 |
| $\lambda$ | 0.5 |
| Learning rate $\eta$ | 0.01 |
| Time constant $\tau$ | 1000 |
| Kernel size | 5 |
| Stride | 1,1 |
| Lateral competition Iterations | 500 |

like blurred, masked, or noisy training samples $D_s$, as shown in Figure 5c Fredrikson et al. (2015); Zhang et al. (2020); Zhao et al. (2021b). To contextualize our SPARSE-GUARD setting, recall that we suppose the attacker has only black-box access to query the model $f_{tar}$ without knowing the details of the target model $f_{tar}$ architecture or parameters like gradient information $\nabla_{Tr}$. The attacker attempts to compute training data reconstruction (i.e., ImRec) attack without having access to other additional information, e.g., blurred or masked images $D_s$.

Two major components of the model inversion attack workflow are the target model $f_{tar}$ and the surrogate attack model $f_{sar}$ Jia & Gong (2018); Dibbo (2023); Zhao et al. (2021a). Training data reconstruction (i.e., ImRec) attack in the literature considers the target model $f_{tar}$ to be either the split network Titcombe et al. (2021) or the end-to-end network Gong et al. (2023); Zhang et al. (2020). In the split network $f_{tar}$ model, the output of a particular layer $l$ in the network, i.e., $a^{[l]}$, where $1 \leqslant l < L$ is accessible to the adversary, whereas, for the end-to-end network, the adversary

Table 6: Experiments set 1 additional Laplace noise benchmark with larger 1.0 noise parameter: Performance in *end-to-end* network setting *(lower rows=better defense)*.

| Dataset | Defense | PSNR ⇊ | SSIM ⇊ | FID ($10^3$) ⇈ | Accuracy |
|---|---|---|---|---|---|
| MNIST | Titcombe (2021)-1.0 | 24.89 | 0.664 | 50.64 | 0.938 |
| | **SPARSE-GUARD0.1** | **19.54** | **0.502** | **178.5** | **0.984** |
| | **SPARSE-GUARD0.25** | **18.81** | **0.340** | **174.1** | **0.983** |
| | **SPARSE-GUARD0.5** | **17.85** | **0.164** | **335.5** | **0.977** |
| Fashion | Titcombe (2021)-1.0 | 20.21 | 0.567 | 80.55 | 0.823 |
| | **SPARSE-GUARD0.1** | **17.92** | **0.209** | **196.1** | **0.897** |
| MNIST | **SPARSE-GUARD0.25** | **17.03** | **0.186** | **195.2** | **0.887** |
| | **SPARSE-GUARD0.5** | **14.51** | **0.069** | **423.2** | **0.876** |

Table 7: Experiments set 2 additional Laplace noise benchmark with larger 1.0 noise parameter: Performance in *split* network setting *(lower rows=better defense)*.

| Dataset | Defense | PSNR ⇊ | SSIM ⇊ | FID ($10^3$) ⇈ | Accuracy |
|---|---|---|---|---|---|
| MNIST | Titcombe (2021)-1.0 | 22.63 | 0.503 | 66.40 | 0.980 |
| | **SPARSE-GUARD0.1** | **16.17** | **0.109** | **227.4** | **0.988** |
| | **SPARSE-GUARD0.25** | **17.40** | **0.058** | **301.6** | **0.980** |
| | **SPARSE-GUARD0.5** | **14.98** | **0.044** | **307.7** | **0.975** |
| Fashion | Titcombe (2021)-1.0 | 18.36 | 0.408 | 80.80 | 0.878 |
| | **SPARSE-GUARD0.1** | **18.11** | **0.154** | **171.1** | **0.904** |
| MNIST | **SPARSE-GUARD0.25** | **17.74** | **0.188** | **203.8** | **0.896** |
| | **SPARSE-GUARD0.5** | **17.15** | **0.134** | **270.4** | **0.879** |

does not have access to intermediate layer outputs; rather, the adversary only has access to the output from the last hidden layer before the classification layer $a^{[L]}$.

# B  ADDITIONAL EXPERIMENTS

In order to try to improve the Laplace noise-based defense, we consider increasing the noise scale parameter $b$ from $\mathcal{L}(\mu=0, b=0.5)$ to $\mathcal{L}(\mu=0, b=1.0)$. Tables 6 and 7 compare these results to SPARSE-GUARD for both datasets in both end-to-end network and split network settings. Observe that the additional noise significantly degrades classification accuracy in all but one case, yet it does not result in reconstruction metrics that rival those of SPARSE-GUARD's. In Figure 6, we present the reconstructed images by different benchmarks along with reconstruction by the Laplace noise-based defense with higher noise parameter $\mathcal{L}(\mu=0, b=1.0)$.

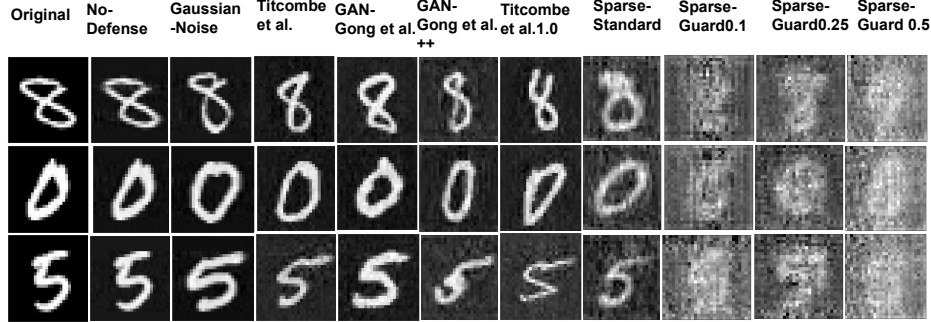

Figure 6: Original images and reconstructed images under SPARSE-GUARD and additional Laplace noise benchmark with larger 1.0 noise parameter.

Table 8: Experiments set 1: Performance in *end-to-end* network setting *(lower rows=better defense)* on CIFAR10 and MedMNIST datasets.

| Dataset | Defense | PSNR ↓↓ | SSIM ↓↓ | FID ($10^3$) ↑↑ | Accuracy |
|---------|---------|---------|---------|---------|----------|
| CIFAR10 | NO-DEFENSE | 21.17 | 0.477 | 70.96 | 0.821 |
| | GAUSSIAN-NOISE | 20.26 | 0.220 | 77.42 | 0.626 |
| | GAN | 19.71 | 0.259 | 132.0 | 0.596 |
| | Titcombe et al. (2021) | 18.62 | 0.174 | 171.9 | 0.792 |
| | Gong et al. (2023)++ | 18.27 | 0.209 | 149.1 | 0.773 |
| | Gong et al. (2023) | 19.10 | 0.150 | 133.8 | 0.682 |
| | SPARSE-STANDARD | 18.01 | 0.003 | 168.6 | 0.790 |
| | **SPARSE-GUARD0.1** | **17.09** | **0.001** | **172.0** | **0.787** |
| | **SPARSE-GUARD0.25** | **16.78** | **0.001** | **189.3** | **0.772** |
| | **SPARSE-GUARD0.5** | **16.24** | **0.001** | **197.0** | **0.744** |
| Medical MNIST | NO-DEFENSE | 31.48 | 0.935 | 10.66 | 0.998 |
| | GAUSSIAN-NOISE | 30.46 | 0.920 | 12.23 | 0.862 |
| | GAN | 27.34 | 0.480 | 33.77 | 0.998 |
| | Gong et al. (2023)++ | 18.37 | 0.353 | 81.52 | 0.894 |
| | Titcombe et al. (2021) | 21.33 | 0.431 | 30.60 | 0.899 |
| | Gong et al. (2023) | 21.52 | 0.436 | 64.88 | 0.770 |
| | SPARSE-STANDARD | 14.79 | 0.119 | 250.6 | 0.907 |
| | **SPARSE-GUARD0.1** | **13.43** | **0.004** | **369.9** | **0.888** |
| | **SPARSE-GUARD0.25** | **12.32** | **0.004** | **375.9** | **0.882** |
| | **SPARSE-GUARD0.5** | **12.04** | **0.004** | **354.1** | **0.881** |

## C  FURTHER SUPPLEMENTARY EXPERIMENTS

We now consider (1) two additional datasets, (2) two additional state-of-the-art defenses, and (3) an additional black-box attack known as Plug and Play (Struppek et al., 2022). In these additional experiments, SPARSE-GUARD outperforms benchmarks by a factor of up to 704. It also has the best PSNR (the most important metric) across every single experiment. We note that in one single experiment, Sparse-Guard has worse SSIM by a 0.001 factor compared to the defense of (Wang et al., 2021b), though it significantly outperforms this defense in terms of PSNR and FID on the same experiment.

### C.0.1  ADDITIONAL DATASETS

We re-run experiments on three additional datasets:

- **CIFAR-10** (Krizhevsky et al., 2009). CIFAR-10 is a high-resolution image dataset with 10 classes, and it allows us to benchmark SYBIL-GUARD on hi-res images;

- **Medical MNIST Larxel (2019).** Medical MNIST is a dataset of real medical images containing six classes (Head CT, Breast MRI, Chest CT, Hand, CXR, and Abdomen CT) that represents a realistic 'worst-case' security application domain.

- **CelebA Liu et al. (2015).** CelebA is a high-resolution celebrity image dataset. It has more than 200K $178 \times 218$ pixel celebrity face images with 40 attribute annotations. *Because this dataset is significantly larger in terms of resolution and image count, compute times for all benchmarks are significantly greater. Therefore, in the interest of time, we compare* SPARSE-GUARD *to the best of the benchmarks, rather than all benchmarks, under end-to-end and Plug and Play settings.*

We present the results on both *end-to-end* and *split* networks in Tables 8 and 9. Also, in Tables 14 and 15, we present comparisons among our SPARSE-GUARD and best performing existing defense (Wang et al., 2021b) in *end-to-end* and *Plug and Play* model inversion attack (Struppek et al., 2022) settings. Finally, we report results on the CelebA dataset in tables 14 and 15. In all of these additional settings, SPARSE-GUARD outperforms all benchmarks.

Table 9: Experiments set 1: Performance in *split* network setting *(lower rows=better defense)* on CIFAR10 and MedMNIST datasets.

| Dataset | Defense | PSNR ↓↓ | SSIM ↓↓ | FID ($10^3$) ↑↑ | Accuracy |
|---------|---------|---------|---------|-----------------|----------|
| CIFAR10 | NO-DEFENSE | 16.48 | 0.709 | 47.77 | 0.823 |
| | GAUSSIAN-NOISE | 14.79 | 0.311 | 149.5 | 0.598 |
| | GAN | 14.87 | 0.296 | 13.01 | 0.675 |
| | Titcombe et al. (2021) | 14.68 | 0.244 | 157.3 | 0.779 |
| | Gong et al. (2023)++ | 13.32 | 0.003 | 162.4 | 0.691 |
| | Gong et al. (2023) | 14.55 | 0.291 | 152.1 | 0.644 |
| | SPARSE-STANDARD | 13.22 | 0.003 | 167.9 | 0.769 |
| | **SPARSE-GUARD0.1** | **13.18** | **0.002** | **174.2** | **0.758** |
| | **SPARSE-GUARD0.25** | **13.07** | **0.002** | **181.2** | **0.742** |
| | **SPARSE-GUARD0.5** | **12.88** | **0.002** | **375.3** | **0.739** |
| Medical MNIST | NO-DEFENSE | 23.47 | 0.776 | 45.57 | 0.993 |
| | GAUSSIAN-NOISE | 21.93 | 0.722 | 44.72 | 0.811 |
| | GAN | 21.67 | 0.719 | 48.49 | 0.912 |
| | Gong et al. (2023)++ | 21.07 | 0.573 | 67.53 | 0.931 |
| | Titcombe et al. (2021) | 21.35 | 0.704 | 48.82 | 0.961 |
| | Gong et al. (2023) | 21.33 | 0.720 | 41.74 | 0.925 |
| | SPARSE-STANDARD | 15.33 | 0.149 | 142.4 | 0.955 |
| | **SPARSE-GUARD0.1** | **13.95** | **0.008** | **244.9** | **0.946** |
| | **SPARSE-GUARD0.25** | **12.31** | **0.008** | **255.3** | **0.928** |
| | **SPARSE-GUARD0.5** | **12.27** | **0.001** | **285.3** | **0.909** |

### C.0.2    ADDITIONAL ATTACKS

We re-run SPARSE-GUARD and all benchmarks under an additional attack setting: the Plug and Play Model Inversion attack (Struppek et al., 2022). We present the performance comparison in Table 12.

### C.0.3    ADDITIONAL DEFENSES

We consider two additional state-of-the-art defenses:

- The very recent differential-privacy DP-SGD defense of Hayes et al. (2023) that is currently under development at Google DeepMind and Meta AI, and is currently the only defense with provable guarantees for model inversion attacks;
- The information regularization-based defense of Wang et al. (2021c).

Our experimental evaluation on all datasets shows SPARSE-GUARD significantly outperforms both defenses in both END-TO-END and SPLIT network settings, as presented in Tables 10 and 11. We also re-run all experiments under the novel Plug and Play Model Inversion attack (Struppek et al., 2022) below.

Table 10: Performance of additional defense benchmarks in *end-to-end* network setting *(lower rows=better defense)*.

| Dataset | Defense | PSNR $\downarrow\downarrow$ | SSIM $\downarrow\downarrow$ | FID ($10^3$) $\uparrow\uparrow$ | Accuracy |
|---------|---------|------|------|------|----------|
| MNIST | Hayes et al. (2023) | 19.75 | 0.488 | 298.8 | 0.871 |
| | Wang et al. (2021b) | 29.05 | 0.817 | 75.39 | 0.985 |
| | SPARSE-STANDARD | 21.34 | 0.439 | 142.9 | 0.986 |
| | **SPARSE-GUARD0.1** | **19.54** | **0.502** | **178.5** | **0.984** |
| | **SPARSE-GUARD0.25** | **18.81** | **0.340** | **174.1** | **0.983** |
| | **SPARSE-GUARD0.5** | **17.85** | **0.164** | **335.5** | **0.977** |
| Fashion MNIST | Hayes et al. (2023) | 21.13 | 0.297 | 223.3 | 0.752 |
| | Wang et al. (2021b) | 25.98 | 0.806 | 41.87 | 0.838 |
| | SPARSE-STANDARD | 19.35 | 0.446 | 128.4 | 0.879 |
| | **SPARSE-GUARD0.1** | **17.92** | **0.209** | **196.1** | **0.897** |
| | **SPARSE-GUARD0.25** | **17.03** | **0.186** | **195.2** | **0.887** |
| | **SPARSE-GUARD0.5** | **14.51** | **0.069** | **423.2** | **0.876** |
| CIFAR10 | Hayes et al. (2023) | 17.95 | 0.002 | 142.4 | 0.626 |
| | Wang et al. (2021b) | 17.08 | 0.002 | 136.1 | 0.793 |
| | SPARSE-STANDARD | 18.01 | 0.003 | 168.6 | 0.790 |
| | **SPARSE-GUARD0.1** | **17.09** | **0.001** | **172.0** | **0.787** |
| | **SPARSE-GUARD0.25** | **16.78** | **0.001** | **189.3** | **0.772** |
| | **SPARSE-GUARD0.5** | **16.24** | **0.001** | **197.0** | **0.744** |
| Medical | Hayes et al. (2023) | 18.48 | 0.007 | 150.9 | 0.824 |
| | Wang et al. (2021b) | 20.48 | 0.549 | 30.01 | 0.986 |
| | SPARSE-STANDARD | 14.79 | 0.119 | 250.6 | 0.907 |
| | **SPARSE-GUARD0.1** | **13.43** | **0.004** | **369.9** | **0.888** |
| | **SPARSE-GUARD0.25** | **12.32** | **0.004** | **375.9** | **0.882** |
| | **SPARSE-GUARD0.5** | **12.04** | **0.004** | **354.1** | **0.881** |

Table 11: Performance of additional defense benchmarks in *split* network setting *(lower rows=better defense)*.

| Dataset | Defense | PSNR $\downarrow\downarrow$ | SSIM $\downarrow\downarrow$ | FID ($10^3$) $\uparrow\uparrow$ | Accuracy |
|---------|---------|------|------|------|----------|
| MNIST | Hayes et al. (2023) | 17.23 | 0.030 | 288.1 | 0.856 |
| | Wang et al. (2021b) | 21.87 | 0.696 | 53.09 | 0.903 |
| | SPARSE-STANDARD | 18.71 | 0.288 | 188.4 | 0.981 |
| | **SPARSE-GUARD0.1** | **16.17** | **0.109** | **227.4** | **0.988** |
| | **SPARSE-GUARD0.25** | **17.40** | **0.058** | **301.6** | **0.980** |
| | **SPARSE-GUARD0.5** | **14.98** | **0.044** | **307.7** | **0.975** |
| Fashion MNIST | Hayes et al. (2023) | 20.10 | 0.256 | 200.6 | 0.748 |
| | Wang et al. (2021b) | 24.53 | 0.588 | 81.79 | 0.881 |
| | SPARSE-STANDARD | 19.54 | 0.405 | 200.5 | 0.882 |
| | **SPARSE-GUARD0.1** | **18.11** | **0.154** | **171.1** | **0.904** |
| | **SPARSE-GUARD0.25** | **17.74** | **0.188** | **203.8** | **0.896** |
| | **SPARSE-GUARD0.5** | **17.15** | **0.134** | **270.4** | **0.879** |
| CIFAR10 | Hayes et al. (2023) | 15.44 | 0.005 | 204.5 | 0.596 |
| | Wang et al. (2021b) | 14.73 | 0.001 | 176.3 | 0.820 |
| | SPARSE-STANDARD | 13.22 | 0.003 | 167.9 | 0.769 |
| | **SPARSE-GUARD0.1** | **13.18** | **0.002** | **174.2** | **0.758** |
| | **SPARSE-GUARD0.25** | **13.07** | **0.002** | **181.2** | **0.742** |
| | **SPARSE-GUARD0.5** | **12.88** | **0.002** | **375.3** | **0.739** |
| Medical MNIST | Hayes et al. (2023) | 21.46 | 0.442 | 137.4 | 0.850 |
| | Wang et al. (2021b) | 20.03 | 0.538 | 65.17 | 0.986 |
| | SPARSE-STANDARD | 15.33 | 0.149 | 142.4 | 0.955 |
| | **SPARSE-GUARD0.1** | **13.95** | **0.008** | **244.9** | **0.946** |
| | **SPARSE-GUARD0.25** | **12.31** | **0.008** | **255.3** | **0.928** |
| | **SPARSE-GUARD0.5** | **12.27** | **0.001** | **285.3** | **0.909** |

Table 12: Performance in Plug and Play Model Inversion Attack (Struppek et al., 2022) setting *(lower rows=better defense)*.

| Dataset | Defense | PSNR ⇓⇓ | SSIM ⇓⇓ | FID ($10^3$) ⇑⇑ | Accuracy |
|---|---|---|---|---|---|
| CIFAR10 | NO-DEFENSE | 11.94 | 0.381 | 39.38 | 0.821 |
| | GAUSSIAN-NOISE | 11.88 | 0.365 | 77.92 | 0.626 |
| | GAN | 11.86 | 0.369 | 88.39 | 0.596 |
| | Titcombe et al. (2021) | 10.89 | 0.346 | 79.19 | 0.792 |
| | Gong et al. (2023)++ | 11.06 | 0.339 | 78.48 | 0.773 |
| | Gong et al. (2023) | 11.21 | 0.334 | 92.33 | 0.682 |
| | SPARSE-STANDARD | 10.74 | 0.303 | 137.4 | 0.790 |
| | **SPARSE-GUARD0.1** | **10.59** | **0.305** | **144.1** | **0.787** |
| | **SPARSE-GUARD0.25** | **10.27** | **0.279** | **189.9** | **0.772** |
| | **SPARSE-GUARD0.5** | **10.23** | **0.276** | **189.7** | **0.744** |
| MNIST | NO-DEFENSE | 7.24 | 0.783 | 23.6 | 0.971 |
| | GAUSSIAN-NOISE | 6.94 | 0.686 | 31.22 | 0.958 |
| | GAN | 6.83 | 0.734 | 89.38 | 0.968 |
| | Gong et al. (2023)++ | 6.69 | 0.716 | 92.21 | 0.987 |
| | Titcombe et al. (2021) | 6.34 | 0.744 | 131.8 | 0.980 |
| | Gong et al. (2023) | 6.76 | 0.681 | 99.53 | 0.985 |
| | SPARSE-STANDARD | 6.24 | 0.631 | 158.6 | 0.986 |
| | **SPARSE-GUARD0.1** | **6.19** | **0.633** | **287.9** | **0.984** |
| | **SPARSE-GUARD0.25** | **5.83** | **0.607** | **289.3** | **0.983** |
| | **SPARSE-GUARD0.5** | **5.74** | **0.604** | **299.6** | **0.977** |
| Fashion MNIST | NO-DEFENSE | 8.91 | 0.147 | 235.5 | 0.886 |
| | GAUSSIAN-NOISE | 8.67 | 0.132 | 239.8 | 0.815 |
| | GAN | 8.66 | 0.147 | 243.3 | 0.883 |
| | Gong et al. (2023)++ | 8.73 | 0.130 | 220.2 | 0.906 |
| | Titcombe et al. (2021) | 8.56 | 0.134 | 229.8 | 0.905 |
| | Gong et al. (2023) | 8.57 | 0.143 | 244.3 | 0.888 |
| | SPARSE-STANDARD | 8.71 | 0.1351 | 223.3 | 0.879 |
| | **SPARSE-GUARD0.1** | **8.49** | **0.039** | **222.8** | **0.897** |
| | **SPARSE-GUARD0.25** | **8.49** | **0.032** | **229.9** | **0.887** |
| | **SPARSE-GUARD0.5** | **8.45** | **0.047** | **233.5** | **0.876** |
| Medical MNIST | NO-DEFENSE | 22.04 | 0.396 | 196.1 | 0.998 |
| | GAUSSIAN-NOISE | 21.83 | 0.382 | 209.4 | 0.862 |
| | GAN | 21.77 | 0.427 | 219.0 | 0.998 |
| | Gong et al. (2023)++ | 21.50 | 0.359 | 273.1 | 0.894 |
| | Titcombe et al. (2021) | 21.68 | 0.360 | 286.3 | 0.899 |
| | Gong et al. (2023) | 21.75 | 0.477 | 249.1 | 0.77 |
| | SPARSE-STANDARD | 20.97 | 0.086 | 239.3 | 0.907 |
| | **SPARSE-GUARD0.1** | **21.19** | **0.057** | **253.5** | **0.888** |
| | **SPARSE-GUARD0.25** | **21.17** | **0.075** | **280.1** | **0.882** |
| | **SPARSE-GUARD0.5** | **20.06** | **0.072** | **288.8** | **0.881** |

Table 13: Performance of additional defense benchmarks in Plug and Play Model Inversion Attack (Struppek et al., 2022) setting.

| Dataset | Defense | PSNR ⇊ | SSIM ⇊ | FID ($10^3$) ⇈ | Accuracy |
|---|---|---|---|---|---|
| CIFAR10 | Hayes et al. (2023) | 11.12 | 0.342 | 142.1 | 0.626 |
| | Wang et al. (2021b) | 11.02 | 0.346 | 142.6 | 0.756 |
| | Sparse-Standard | 10.74 | 0.303 | 137.4 | 0.790 |
| | **Sparse-Guard0.1** | **10.59** | **0.305** | **144.1** | **0.787** |
| | **Sparse-Guard0.25** | **10.27** | **0.279** | **189.9** | **0.772** |
| | **Sparse-Guard0.5** | **10.23** | **0.276** | **189.7** | **0.744** |
| MNIST | Hayes et al. (2023) | 7.03 | 0.672 | 396.1 | 0.871 |
| | Wang et al. (2021b) | 7.14 | 0.752 | 261.2 | 0.937 |
| | Sparse-Standard | 6.24 | 0.631 | 158.6 | 0.986 |
| | **Sparse-Guard0.1** | **6.19** | **0.633** | **287.9** | **0.984** |
| | **Sparse-Guard0.25** | **5.83** | **0.607** | **289.3** | **0.983** |
| | **Sparse-Guard0.5** | **5.74** | **0.604** | **299.6** | **0.977** |
| Fashion MNIST | Hayes et al. (2023) | 8.63 | 0.139 | 218.4 | 0.752 |
| | Wang et al. (2021b) | 8.90 | 0.119 | 210.3 | 0.88 |
| | Sparse-Standard | 8.71 | 0.1351 | 223.3 | 0.879 |
| | **Sparse-Guard0.1** | **8.49** | **0.039** | **222.8** | **0.897** |
| | **Sparse-Guard0.25** | **8.49** | **0.032** | **229.9** | **0.887** |
| | **Sparse-Guard0.5** | **8.45** | **0.047** | **233.5** | **0.876** |
| Medical MNIST | Hayes et al. (2023) | 21.72 | 0.337 | 259.7 | 0.823 |
| | Wang et al. (2021b) | 21.71 | 0.322 | 211.7 | 0.937 |
| | Sparse-Standard | 20.97 | 0.086 | 239.3 | 0.907 |
| | **Sparse-Guard0.1** | **21.19** | **0.057** | **253.5** | **0.888** |
| | **Sparse-Guard0.25** | **21.17** | **0.075** | **280.1** | **0.882** |
| | **Sparse-Guard0.5** | **20.06** | **0.072** | **288.8** | **0.881** |

Table 14: **CelebA Results:** Performance comparison with the best defense Wang et al. (2021b) in *end-to-end* network setting *(lower rows=better defense)* on high resolution CelebA dataset.

| Dataset | Defense | PSNR ⇊ | SSIM ⇊ | FID ($10^3$) ⇈ | Accuracy |
|---|---|---|---|---|---|
| CelebA | No-Defense | 16.26 | 0.262 | 201.8 | 0.773 |
| | Wang et al. (2021b) | 13.63 | 0.001 | 203.2 | 0.744 |
| | Sparse-Standard | 13.09 | 0.003 | 222.1 | 0.749 |
| | **Sparse-Guard0.1** | **12.89** | **0.004** | **228.5** | **0.748** |
| | **Sparse-Guard0.25** | **12.73** | **0.004** | **218.9** | **0.737** |
| | **Sparse-Guard0.5** | **12.72** | **0.002** | **231.9** | **0.742** |

Table 15: **CelebA Results:** Performance comparison with the best defense Wang et al. (2021b) under the **Plug and Play** Model Inversion Attack (Struppek et al., 2022) setting *(lower rows=better defense)* on high resolution CelebA dataset.

| Dataset | Defense | PSNR ⇊ | SSIM ⇊ | FID ($10^3$) ⇈ | Accuracy |
|---|---|---|---|---|---|
| CelebA | No-Defense | 8.51 | 0.196 | 78.58 | 0.779 |
| | Wang et al. (2021b) | 7.93 | 0.165 | 80.55 | 0.742 |
| | Sparse-Standard | 7.81 | 0.159 | 81.34 | 0.728 |
| | **Sparse-Guard0.1** | **7.29** | **0.138** | **181.4** | **0.726** |
| | **Sparse-Guard0.25** | **6.62** | **0.092** | **180.5** | **0.739** |
| | **Sparse-Guard0.5** | **6.57** | **0.107** | **184.0** | **0.723** |

Figure 7: Distributions of reconstructed images' SSIM after attacking the NO-DEFENSE network in end-to-end and Plug and Play settings on the **CelebA** dataset. Note that on this NO-DEFENSE network, the attacks achieve almost perfect reconstruction on small but significant number of images in both settings (mass on the right of the histograms).

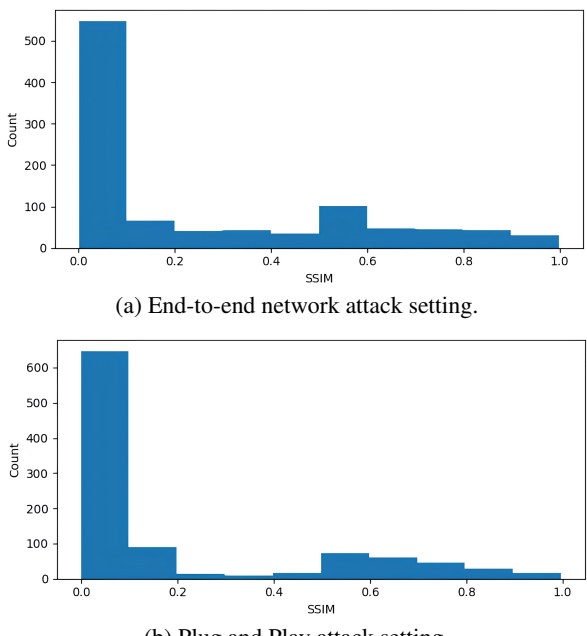

(a) End-to-end network attack setting.

(b) Plug and Play attack setting.

Table 16: We note that our basic SPARSE-GUARD research implementation completes in comparable or less compute time than highly optimized implementations of benchmarks. In the 'worst-case' across all of our experiments, SPARSE-GUARD is slightly slower than benchmarks – **we reprint the compute times (in seconds) for this 'worst-case' experiment below (The MNIST dataset under the Plug and Play attack (Struppek et al., 2022)).**

| *Model* | TIME (SEC) |
|---|---|
| NO-DEFENSE | 10555.3 |
| GAUSSIAN-NOISE | 12555.3 |
| GAN | 15762.4 |
| Titcombe et al. (2021) | 14390.2 |
| Gong et al. (2023) | 16061.8 |
| Gong et al. (2023)++ | 17521.8 |
| Hayes et al. (2023) | 16923.9 |
| Wang et al. (2021b) | 15229.9 |
| SPARSE-STANDARD | 12327.5 |
| SPARSE-GUARD0.1 | 17009.8 |
| SPARSE-GUARD0.25 | 17181.2 |
| SPARSE-GUARD0.5 | 17912.9 |

