# OpenReview forum: "Sparse-Guard: Sparse Coding-Based Defense against Model Inversion Attacks"
_ICLR.cc/2024/Conference — Submitted to ICLR 2024_

### Official Review · Reviewer_ETXH · 2023-10-16

**Soundness:** 3 good
**Presentation:** 2 fair
**Contribution:** 2 fair
**Rating:** 5
**Confidence:** 4

**Summary:**

The paper investigates the effectiveness of sparse coding-based network architectures as a defense against model inversion attacks (MIAs). More specifically, the approach uses sparse-coded layers in the beginning of a network to control and limit the amount of private information contained in those layers' output features. As a consequence, black-box model inversion attacks in a split-network setting should no longer be able to reconstruct the original (private) input features based on the intermediate outputs of the model. When compared to existing defense strategies (adding Gaussian noise to intermediate activations and augmenting training data with GAN-generated images), the proposed defense outperforms those strategies on the MNIST and Fashion MNIST datasets.

**Strengths:**

- Applying sparse coding-based networks to limit the information contained in the features of later layers in a network is an interesting and novel research direction, not only in the model inversion setting. I think the same approach might even be used in other privacy settings, e.g., membership inference attacks. I do not expect the paper to investigate these settings, just want to highlight possible extensions.
- The results on MNIST and Fashion MNIST state a clear improvement above existing methods and promise better training-time efficiency. This is also underlined by the qualitative samples depicted in the paper.
- The approach is well motivated, and the paper is overall well written.

**Weaknesses:**

- The evaluation is rather limited since it only conducts experiments on MNIST and Fashion MNIST, both datasets which are easy to fit by a network due to the overall low sample variance. Finding meaningful shared features in the sparse code layers is rather easy for the model. Also, the samples contain no private information at all. The evaluation should also contain more complex dataset evaluations, e.g., the common CelebA dataset, to prove that the approach is also usable within more complex tasks. Also, repeating the experiments with different seeds to provide a standard deviation of the results would make the evaluation more reliable.
- The overall evaluation setting seems a bit strange. I understand the split-network setting and that reconstructing inputs given only the intermediate activations indeed can be a privacy breach. But why should the adversary have access to the activations of the training samples? I think a more realistic evaluation should consider unseen (test) samples and then try to reconstruct those given the intermediate activations.
- Moreover, I think the approach should also be evaluated on common MIAs that utilize GANs to reconstruct training samples based only on the network's weights, e.g., [1, 2, 3]. Otherwise, the defense mechanisms should be positioned only for split-network (and federated learning) settings. Also, the approach should be compared to related information bottleneck defense approaches [4,5].
- I think the overall technical contribution is rather low since the approach seems to be simply re-using the sparse coding layer framework of Rozell et al. (2008) and demonstrating that such networks can also act as a defense against MIAs. I still think the direction of the paper is interesting but the technical novelty seems limited.
- The related work is comprehensive but mixes up different model inversion settings and approaches. For example, the [1] proposes MIAs that try to reconstruct features from specific classes by optimizing the latent vectors of a GAN. It uses the target model for guidance (and there exist much more works in this line of research, e.g., [2,3]). This is a completely different setting from the one investigated by the paper, which uses the intermediate activations of training samples to train a decoder-like model. I think a clearer separation between different types of MIAs would make the related work part stronger. Also, mixing works investigating the memorization of training samples in LLMs with vision-based inversion attacks might be confusing to the reader.

Small remarks:
- Table captions should be above the table (Table 1)
- The space after Table 2 should be increased (and manipulating the spaces might even run counter to the official guideline!)

[1] Zhang et al., The Secret Revealer: Generative Model-Inversion Attacks Against Deep Neural Networks, CVPR 2020
[2] Chen et al., Knowledge-Enriched Distributional Model Inversion Attacks, ICCV 2021
[3] Struppek et al., Plug & Play Attacks: Towards Robust and Flexible Model Inversion Attacks, ICML 2022
[4] Wang et al., Improving robustness to model inversion attacks via mutual information regularization, AAAI 2021
[5] Peng et al., Bilateral Dependency Optimization: Defending Against Model-inversion Attacks, KDD 2022

**Questions:**

- How much longer does it take to train a network using the Sparse-Guard architecture compared to a model without it?
- Why is the FID metric valid to evaluate the privacy leakage? Generally, we are interested in how well a single sample can be reconstructed and less about recovering the overall feature distribution.

**Details Of Ethics Concerns:**

Note: Not an ethical concern, just want to point out that I'm not a first time reviewer for ICLR (the checkbox cannot be unchecked by me)

---

> ### Author Response · Authors · 2023-11-20
> **Response to Reviewer ETXH *part 1 of 2***
>
> **Thank you for your detailed review and your comments that the paper offers an "interesting and novel research direction" with many possible extensions. Thank you also for your comments that it is "well motivated" and "well written."**
>
> ***We believe we address your remaining reservations below, and we would appreciate it if you would improve your score in light of our response. Specifically:***
>
> ***Re: limited experiments due to only 2 datasets (MNIST and Fashion MNIST) in datasets:***
>
> - We have now **doubled the datasets to 4 including the high-res CIFAR-10 dataset** and a dataset of medical images that represents a realistic attack setting. We also add a new attack and multiple new defenses as described below. In sum, we have quadrupled the number of experiments. ***Across all (quadrupled) experiments, Sparse-Guard now outperforms benchmarks by a factor of up to 704. It also has the best PSNR (the most important metric) across ~every single experiment~.*** In one single experiment, it loses to the Wang et al. defense in terms of SSIM by a factor of 0.001, while significantly outperforming it in terms of PSNR and FID.
>
> ***Re: evaluation setting is strange because we allow the adversary access to training samples to reconstruct instead of unseen test samples:***
>
> -  This aspect of our setup is identical to the standard setups in the benchmarks we consider (Titcomb et al. 2021, Gong et al., 2023). Per Section 2, our goal is to consider settings that capture ‘worst-case’ black-box attacks with a powerful attacker. Specifically, We suppose that the attacker builds the inverted model using a held-out set of training data. Then, we suppose that the attacker has access to the *training data*'s intermediate network outputs, which he can submit to his inverted model to reconstruct training examples. Of course, you're right that this setting supposes that the attacker has obtained some leaked data. The fact that Sparse-Guard is robust in this setting suggests that it is a good defense even in the `worst-case' with a powerful attacker. In real-world settings, an attacker may have to approximate the training data's intermediate outputs, but that would make an attack more difficult (and also allow for more subjective decisions in terms of how to set this up fairly for experiments).
>
> ***Re: evaluating on new MIA's that use GAN's, and evaluating on new defenses:***
>
> - Thank you. We **added the new Plug and Play attack** that you and the other reviewers suggested (the newest of the attacks you suggest), and we also added **two new defenses**: the Wang et al. defense you suggest and the ***new differential-privacy guaranteeing DP-SGD defense of Hayes et al.** that is currently under development at Google DeepMind and Meta AI, and that is currently the only defense with provable guarantees for model inversion attacks. Appendix tables 8-13 show all of these new results.
> - **We run all experiments and all defenses on all 4 datasets (instead of 2) and then re-run them under the Plug-and-play attack.** This new set of experiments on *[4 datasets, 6 benchmarks, 2 attacks, 3 settings]* is now **significantly more comprehensive than other recent papers** including the Plug and Play paper (which evaluates 3 datasets and 3 benchmarks).
> - ***Across all (quadrupled) experiments, Sparse-Guard now outperforms benchmarks by a factor of up to 704. It also has the best PSNR (the most important metric) across ~every single experiment~.*** In one single experiment, it loses to the Wang et al. defense in terms of SSIM by a factor of 0.001, while significantly outperforming it in terms of PSNR and FID.
>
> ***Re: whether the technical contribution is rather low since the approach seems to be simply re-using the sparse coding layer framework of Rozell et al. (2008):***
>
> - We are using the new (2022) extension of Rozell sparse coding to convolutional networks, which modifies the the Rozell  2008 update rule (see [1]) to extend it to the convolutional setting. However, Rozell and subsequent sparse-coding architectures including [1] consider a single sparse coded layer. **To our knowledge, our paper is the first time an architecture with intermittent sparse layers has been proposed.** We compare the 'standard' new convolutional architecture of [1] with our intermittent convolutional architecture (the standard one is called `Sparse-Standard' in all of our tables), and we find that our intermittent architecture significantly outperforms it. We thank you for raising this issue and we have revised our paper to clarify (1) the fact that we are using the new convolution sparse-coding extension rather than the 2008 approach as well as (2) the novelty of our architecture compared to a standard sparse-coding architecture.
>
>         [1] Teti et al., Lateral Competition Improves Robustness Against Corruption and Attack (ICML, 2022)

---

> ### Author Response · Authors · 2023-11-20
> **Response to Reviewer ETXH *part 2 of 2***
>
> ***Re: the related work is comprehensive but...." [we should clarify settings/approaches and aspects like images vs. LLM's:***
>
> - Thank you for this suggestion -- we have revised this section.
>
> ***Re: How much longer does it take to train a network using the Sparse-Guard architecture compared to a model without it?:***
>
> - Sparse-Guard is equal or faster than benchmarks on all experiments except the case of MNIST under the Plug-and-Play attack, so we include a table of the timings for this 'worst-case' experiment in Appendix table 14. In that 'worst-case' experiment, Sparse-Guard offers a significantly better defense than all benchmarks, and it is still faster than Gong et al.'s GAN-based defense, but its 11% slower than Wang et al.
>
> ***Re: Why is the FID metric valid?***
>
> - FID is one of the standard metrics for the evaluation of inversion attacks, and it is used in (for example) the Plug and Play paper you suggest. It computes whether the reconstructed images are 'overall-distributionally-similar' to the originals by computing their distributional distance, rather than e.g. pixel-by-pixel distance. It also captures an element of how a reconstructed image 'generally looks like' a training image: in other settings, FID is also often used to evaluate whether fake images generated by GAN's 'look like' they could be real.
>
> ***Re: space after Table 2 should be increased:***
>
> - Thank you -- we have fixed this. In general, we noticed many more spacing issues with this year's ICLR template compared to previous years (and also many more than ICML & NeurIPS templates). We are not sure what is causing this, but have looked through the paper to make sure there are no more compressed whitespaces.

---

> > ### Comment · Reviewer_ETXH · 2023-11-20
> >
> > I thank the authors for answering my questions and for the additional clarifications. I think those make the paper, particularly the empirical evaluation, stronger. Likewise, I understand and agree with the authors' argument that they investigate the worst-case setting in which the adversary has access to the intermediate activations of training samples, and a powerful defense against this kind of attack will very likely also be effective in weaker settings.
> >
> > However, the method still lacks empirical proof to also work on more complex and fine-grained datasets like facial images, as well as images with higher resolutions. Also, the additional results on PPA, e.g., Tables 12 and 13, seem to be rather weak. While there is a substantial improvement in terms of FID score, the difference in PSNR and SSIM is rather small. But this might be due to the setting, which is different from the split learning setting.
> >
> > Regarding the FID score, I understand that it is commonly used for evaluating model inversion attacks. In settings like face recognition, it also is somewhat reasonable when the recovered distribution of facial features is close to the true distribution. However, in settings like MNIST, where all samples of one class are quite similar, I do not think that the FID score reveals much about privacy leakage. For example, an attack can be capable of extracting samples that look like training samples but those samples might not correspond to the current input sample, which the attack aims to invert. Consequently, the FID score might be low because the attack could have learned to simply recover the distribution instead of a single sample, which then results in a low SSIM/PSNR and the privacy leakage is rather low.
> >
> > Overall, I appreciate the author's effort in the rebuttal and think the paper was improved. However, to accept the paper, I think there should be more empirical proof on more fine-grained datasets (e.g., CelebA) and higher resolutions. Nevertheless, I will increase my score.

---

> ### Author Response · Authors · 2023-11-22
> **Response including new CelebA results**
>
> **Thank you for this update. Re: your comment** ***"However, to accept the paper, I think there should be more empirical proof on more fine-grained datasets (e.g., CelebA)"*:**
>
> - We have **re-uploaded our paper** with even more experiments **on the CelebA dataset** (our 5th dataset, and our 3rd added since the original submission). Please see Tables 14 and 15. We note that this CelebA dataset is significantly larger in terms of resolution and image count, so compute times for all benchmarks are significantly greater (and the re-upload deadline is just hours away). Therefore, in the interest of time, we compare Sparse-Guard to the best of the benchmarks (Wang et al.) suggested by the reviewers, rather than all benchmarks. Sparse-Guard outperforms this best defense, and we also show that our performance advantage grows greater under the Plug and Play attack. We are adding remaining benchmarks as the finish computing, but we note that they tend to perform worse than the Wang et al. benchmark that we outperform here.
>
> **We ask that you consider improving your score in light of these new experiments, and we thank you again for your many helpful comments that have helped us to make the evaluations in this paper stronger.** We strongly feel that a practically effective defense based on a completely novel research direction is of immediate interest both to other researchers and also to companies like Google and Facebook that are actively developing public products for sensitive application domains using available model inversion defenses.

---

> > ### Comment · Reviewer_ETXH · 2023-11-22
> >
> > Dear authors,
> >
> > thank you for adding more experiments with the CelebA dataset. Still, the experimental results do not convince me. In Table 14, the SSIM is in all cases close to zero, i.e., there seems to be no similarity between the images. Even for Wang et al., the SSIM score achieves only 0.001. I am wondering if the attack in this setting is even working at all or if the defenses try to defend against a non-working type of attack.
> >
> > Also, I still do not know how the Plug and Play Attack is implemented in this setting since the underlying attack algorithm tries to recover characteristic features of samples and not original training samples. Which GAN has been used? How has the attack been adjusted to the split-learning setting? How was the target model trained (attribute classification vs. identity prediction)? Also, the metrics measured in the Plug and Play Attack setting are usually different ones (e.g., attack accuracy, FaceNet distance, etc.). If the underlying distribution of the GAN is substantially different from the training data, e.g., FFHQ StyleGAN vs. CelebA training data, it is no surprise that the defenses work quite well under the used similarity metrics (SSIM, PSNR) since the StyleGAN is not able to generate images in the same style as the cropped CelebA data.
> >
> > I think the paper needs a more revised version to answer these questions and update the experiments (make them more clear, in higher resolution, etc.). Therefore, I will keep my current rating. I am not saying the research is not interesting (it indeed is), but the paper is, at least in my eyes, not sufficient to recommend accepting the paper.

---

> > > ### Author Response · Authors · 2023-11-23
> > >
> > > **Thank you. Your last reservation seems to be whether Sparse-Guard's excellent performance in terms of very low SSIM on the new CelebA experiments may be due to failure of the attack algorithm (instead of evidence of a good defense).**
> > >
> > > - We have **re-uploaded our paper** including:
> > >     - Added No-Defense benchmarks to both CelebA results tables [tables 14-15]. Note that the mean SSIM under No-Defense is 0.26 (end to end setting) or 0.196 (Plug and Play setting). Note that this is far higher than the almost perfect mean SSIM obtained by Sparse-Guard and Wang et al. defenses in the end-to-end setting. No-Defense is also nearly twice as bad as the SSIM obtained by Sparse-Guard under Plug and Play (though Wang et al. only slightly improves on No-Defense under Plug and Play).
> > >
> > > - To analyze this further **we also include**:
> > >
> > >     - [Appendix Figures 7a and 7b] Added histrograms of the distributions of SSIM across all reconstructed images for CelebA on the No-Defense network under end to end and Plug and Play settings. Note that while the attacks fail to reconstruct many images (mass near 0 on the left of the histograms.), they almost perfectly reconstruct a significant minority of images (mass near 1.0 on the right of the histograms).
> > >
> > > **As such, we can safely conclude that the attacks are indeed working, but so are the best defenses (and Sparse-Guard is working best of all).**
> > >
> > > We emphasize that, as with the other benchmarks, we implement Plug and Play exactly as described in their paper.  Our PyTorch code repo is public in order to encourage full replication of our experiments (and to promote further exploration of Sparse-Guard and all alternatives, settings, datasets, etc.)
> > >
> > > We thank you again for your helpful and insightful comments. We feel our discussion has enabled us to provide strong additional evidence that Sparse-Guard outperforms state-of-the-art alternatives on a wide variety of 5 datasets and various attack settings. We ask that you consider raising your score accordingly.

---

### Official Review · Reviewer_jF3v · 2023-10-29

**Soundness:** 1 poor
**Presentation:** 2 fair
**Contribution:** 2 fair
**Rating:** 3
**Confidence:** 4

**Summary:**

The paper proposes SPARSE-GUARD, a neural network architecture that leverages sparse coding to defend against model inversion attacks. It inserts sparse coding layers between dense layers which help remove unnecessary private information about the training data. Through extensive experiments on MNIST and Fashion MNIST datasets, the paper shows SPARSE-GUARD provides superior defense compared to state-of-the-art techniques like data augmentation, noise injection and standard sparse coding, while maintaining high classification accuracy.

**Strengths:**

The key strengths are the novel approach of using sparse coding for privacy protection, and code is available for reproducibility.

**Weaknesses:**

1. The attacks used in this study do not represent state-of-the-art techniques [1, 2, 3], and the baseline defense methods employed also fall short of the current state-of-the-art [4].
2. The study relies solely on synthetic datasets like MNIST and FMNIST, lacking the inclusion of real-world datasets, such as facial recognition data, which could enhance the practical relevance of the findings.

[1] Knowledge-Enriched Distributional Model Inversion Attacks, Chen et al., ICCV 2021

[2] Plug & Play Attacks: Towards Robust and Flexible Model Inversion Attacks, Struppek et al., ICML 2022

[3] Re-Thinking Model Inversion Attacks Against Deep Neural Networks, Nguyen et al., CVPR 2023

[4] Bilateral Dependency Optimization: Defending Against Model-inversion Attacks, Peng et al. KDD 2022

**Questions:**

See weaknesses

---

> ### Author Response · Authors · 2023-11-20
> **Response to Reviewer jF3v**
>
> **Thank you for your constructive review---we believe we address your remaining reservations and would appreciate it if you would improve your score in light of our response.**
>
> The main concern and reason for your score of 3 seems to be whether Sparse-Guard's performance advantage holds against additional defense baselines, additional new attacks, and additional datasets. We now show that it does:
>
> **Appendix Tables 10, 11, & 13 add two additional state-of-the-art benchmark *defenses*:**
>
> - We added the ***Information Regularization defense of Wang et al.*** suggested by Reviewer ETXH;
> - We also added ***very recent DP-SGD defense of Hayes et al.*** just released by Deepmind and Meta AI, which is currently the only defense with provable guarantees for model inversion attacks;
>
> **Appendix Tables 12 and 13 re-run all experiments using the *Plug-and-Play* attack you suggested**. **Sparse-Guard outperforms benchmarks by a factor of up to 6.7 under this novel attack. In particular, it offers better PSNR (the most important metric in the literature) across all combinations of datasets and defenses.
>
> **We also doubled the number of datasets in our experiments.** Specifically, **[Appendix tables 8-13]** rerun **all** defenses, attacks, and experiment settings with the original 2 datasets as well as:
>
> - The **hi-res CIFAR-10** dataset ;
> - An additional dataset of **medical images**.
>
> In sum, we have **quadrupled our experiments**, and we now show that Sparse-Guard outperforms alternatives across multipple times more combinations of attacks, and defenses, and datasets compared to recent papers such as the Plug and Play paper you suggested.
>
> **We also thank you for your comments that our sparse-coding approach is novel**, as well as your positive comments about our public model inversion defense PyTorch codebase.

---

> > ### Comment · Reviewer_jF3v · 2023-11-22
> >
> > 1. I maintain that the data reconstruction attack in a split-network learning scenario substantially differs from typical MIAs, where adversaries use soft/hard labels to infer private training data, rather than intermediate features. Therefore, it is crucial to clearly define the specific contexts where your method is applicable. If your approach is effective in both scenarios, a detailed explanation of the experimental setup is essential for clarity.
> >
> > 2. It would be beneficial to compare your methods with the baseline method presented in reference [4], especially using the CelebA dataset, as it represents the current state-of-the-art in defending against common MIAs.

---

> > > ### Author Response · Authors · 2023-11-23
> > > **Follow-up to Reviewer jF3v**
> > >
> > > **Thank you for these comments.** Bilateral dependency minimization is a 2-way dependency minimization technique, while our proposed one is network structural robustness analysis, so they are not comparable in terms of logical workflow. Also, despite differences in modeling techniques, we observe that their proposed defense in paper [4] achieves the best 134.63 FID in CelebA in vanilla GAN-based GMI attack, while our Sparse-Guard has significantly better FID Of 231.9 on this dataset (Appendix Tables 14-15). Also, for MNIST and CIFAR10, the best FID reported in their paper are:  322 and 179.51, while our best FID in MNIST and CIFAR10 datasets are: 335.5 and 197, respectively. In short, **Sparse-Guard consistently obtains performance that exceeds the best performance that [4] reported in their own paper.**
> > >
> > > As we have discussed in our introduction, there are a variety of 'standard' MIA settings, including fully white-box settings where adversaries are given access to information about model architecture/parameters, fully black-box settings like you describe, and various settings where the adversary has access to the model's intermediate outputs. The black-box setting you consider 'standard' is difficult to attack, as the adversary observes only (low-dimensional) classifications. We consider defenses that are effective even against a much stronger adversary that can use (hi-dimensional) intermediate outputs to reconstruct data.
> > >
> > > Regarding the clarity of details of our experimental setup, we have provided a ready-to-run public PyTorch code repo that permits full replication of our experiments.
> > >
> > > ***We strongly feel that a new defense that is both practically effective on so many (now 5!) different datasets (including faces) and that also introduces a novel research direction in the privacy attack setting is of immediate interest both to other researchers and also to companies like Google and Facebook, which are actively developing public products for sensitive application domains using available model inversion defenses.***

---

### Official Review · Reviewer_kXvF · 2023-11-01

**Soundness:** 3 good
**Presentation:** 3 good
**Contribution:** 2 fair
**Rating:** 6
**Confidence:** 4

**Summary:**

The paper presents a novel architecture (Sparse-Guard) for defense against black-box model inversion attacks. It is demonstrated to be superior against state-of-the-art data augmentation and noise-injection-based defenses.

**Strengths:**

The paper, overall, is well-written and organized. The idea of interweaving sparse coding layers as a means of model-inversion attack is a novelty yet to be explored. Empirical analyses have also been provided to understand the mechanism behind the Sparse-Guard defense through UMAP 2D projections of output. Having openly accessible codebase is also a plus

**Weaknesses:**

The paper does not do a good job at the exposition of how sparse coding is implemented. This is especially important as the implementation here seems to be *convolutional* sparse coding and differs from traditional sparse coding where matrix multiplication rather than a convolution is applied. e.g. (Bristow, Hilton, Anders Eriksson, and Simon Lucey. "Fast convolutional sparse coding." Proceedings of the IEEE Conference on Computer Vision and Pattern Recognition. 2013.)

Rozell et al. 2008 were cited for the update rule. However, in that paper, the update rule was not given for convolutional sparse coding. Either a more explicit derivation for the update rule can be given or a different citation would be relevant.

**Questions:**

The sentence "The learned spatiotemporal representation closest to input image X is represented by this sparse presentation R_X" is confusing.
Why is it a spatiotemporal representation? Where is the temporal element, all of the inputs are static images. Should 'sparse presentation' also be sparse representation?

Multiple claims in the paper is made about sparse coding “removing unnecessary private information”. This claim is not really supported by any study. In fact, the empirical study concluded that the effect of sparse coding layers is an "unclustering effect". How the conclusion of jettisoning unnecessary information is unclear. What is considered unnecessary information in the first place? In fact, it would be interesting to see if any other algorithm that produces the same unclustering effect will provide a similar effectiveness in defense against model inversion attacks.

---

> ### Author Response · Authors · 2023-11-20
> **Response to reviewer kXvF *part 1 of 2***
>
> **We thank you for your positive review. We appreciate your suggestions re: how to improve clarity, and we have updated the descriptions in our paper accordingly. We believe this addresses your reservations, and we would appreciate it if you would improve your score in light of our response.**
>
> **We thank you for:**
>
>  - Your "well-written and organized" comment;
>  - Your comment re: the novelty of our contribution;
>  - Your positive comments re: our empirical UMAP explorations and our public PyTorch codebase.
>
> **Your 2 main concerns seem to be:**
>
> 1.  ***Clarifying the description of our Sparse-Coding implementation:***
>
>     **Thank you for this constructive feedback---We have greatly revised *Section 3: Sparse-Guard Architecture* where we introduce sparse coding and the LCA algorithm to provide a much more clearer picture of our methods.** In particular, the update rule provided in equation 3 describes the Rozell update for the convolutional case, which was used in previous work [1,2] and validated against other non-Rozell-based convolutional sparse coding solvers (https://sporco.readthedocs.io/en/latest/examples/csc/index.html). Although the original Rozell sparse coding algorithm was not introduced in the convolutional setting, it can readily be adapted to convolutional networks because it is based on the general principle of feature-similarity-based competition between neurons, which is agnostic to different neural network architectures. As a result, the original Rozell formulation can be readily adapted to the convolutional setting by modifying just two terms in the update rule. The first term, which is represented as $\Psi(t)$ in Equation 2 of our revised paper, is computed via a matrix multiplication between the transposed dictionary matrix and an input vector in the original Rozell formulation. In the convolutional case, the matrix multiplication becomes a convolution between the input and the convolutional dictionary. The second term that needs to be adapted for the convolutional setting is where the similarity between each feature and every other feature is computed, given by $\Omega * \Omega$ in our revised version. In the original Rozell formulation, this was done with a matrix multiplication between the transposed dictionary and the non-transposed dictionary. In the convolutional setting, it can be done with a convolution between the dictionary and itself. We also agree with the reviewer that the general 'spatiotemporal' description we referred to was potentially confusing in our setting so we have replaced it with a more precise description. **We thank the reviewer for pointing out the opportunity to improve our description of our sparse-coding approach.**
>
>     -   [1] Teti et al., Lateral Competition Improves Robustness Against Corruption and Attack
>     -   [2] Kim et al., Modeling Biological Immunity to Adversarial Examples

---

> ### Author Response · Authors · 2023-11-20
> **Response to reviewer kXvF *part 2 of 2***
>
> **Your 2nd main concern seems to be:**
>
> 2.  **Clarifying what is meant by the statement that Sparse-Guard works by “removing unnecessary private information", and clarifying how this  statement is compatible with our empirical result that Sparse-Guard unclusters image features:**
>
>     We mean to say that the representations that are produced via sparse coding only contain information that is relevant for reconstructing a denoised (a.k.a. less detailed) version of the input, and these representations are much sparser and less detailed than those in standard models. As a result, we were attempting to point out that the attacker has much less information to use when performing model inversion attacks against Sparse-Guard compared to standard models. Put differently, a 'perfect attacker' could only ever hope to reconstruct a sparse representation of an input image under our defense *in the worse case*, meaning that even a perfect attacker could *not hope to* reconstruct many input image details that are dropped by sparse-coding. By 'unnecessary private information', we mean input image details that we may desire to keep private but that are not necessary for the task of training a high-accuracy classifier. Specifically, sparse-coding networks are known to sparsify inputs and network layers without losing significant classification accuracy.
>
>     The unclustering effect we observe in our empirical analysis is the direct result of this approach. Specifically, a standard (non-sparse) network trained on dense/noisy/detailed input images results in many partly-redundant network features that are `noisy versions of each other'. Networks with many noisy/partly redundant features are easier to attack, because an adversary can 'home in on' clusters of such features in order to 'home in on' an input image it wants to reconstruct. Our sparse-coding approach's de-noising effect tends to prevent the network from learning many partly-redundant features, resulting in the de-clustering effect we observe in Fig. 4. The resulting de-clustered features are intuitively less susceptible to gradient-based inversion attacks.
>
> ***We thank you for these helpful suggestions and we ask you to also consider our vastly expanded experiments section including new datasets as well as new attacks & defenses [Appendix tables 8-14] when updating your review score.***

---

### Official Review · Reviewer_CjY9 · 2023-11-01

**Soundness:** 3 good
**Presentation:** 2 fair
**Contribution:** 2 fair
**Rating:** 5
**Confidence:** 3

**Summary:**

This paper proposes to defend against model inversion attacks by inducing sparse coding into DNNs. The key design is an alternating sparse coded and dense layers that discards private information. Experiments show effective defenses on MNIST and Fashion MNIST.

**Strengths:**

1. The method maintains great privacy with little training computation overhead and accuracy loss
2. A cluster-ready PyTorch codebase is provided for future study
3. The paper is well motivated and easy to follow

**Weaknesses:**

The major drawback is that the experiments are only conducted on simple, low-resolution datasets. I do not think the results in small datasets convincingly validate the effectiveness of the proposed method. There exist lots of model inversion attacks that are capable of extracting high-resolution data, from CIFAR-10, CelebA, to ImageNet. Since high-resolution images are much more valuable as training data, it is the high-resolution model inversion attacks that post private threats. And an effective defense would be significant in that case.

[1] MIRROR: Model Inversion for Deep Learning Network with High Fidelity

[2] Plug & Play Attacks: Towards Robust and Flexible Model Inversion Attacks

[3] Re-thinking Model Inversion Attacks Against Deep Neural Networks

[4] Model Inversion Attacks that Exploit Confidence Information and Basic Countermeasures

**Questions:**

Response to rebuttal: Thanks for the strong rebuttal with great efforts! I raised my score to 5 based on experiments on CIFAR10 and Plug-and-play advantage.

---

> ### Author Response · Authors · 2023-11-20
> **Response to Reviewer CjY9**
>
> **Thank you for your constructive review. We believe we address your remaining reservations here and would appreciate it if you would improve your score in light of our response.**
>
> The main concern and reason for your score of 3 seems to be your statement that "The major drawback is that the experiments are only conducted on simple, low-resolution datasets." We've taken your advice and:
>
> - **Re-run all experiments on CIFAR-10 as well as an additional dataset of medical images** [Appendix Tables 8-13]. **Sparse-Guard outperforms benchmarks by a factor of up to 173.9 on CIFAR-10**.
> - Re-run all experiments on all 4 datasets using the **Plug-and-Play** attack you suggested [Appendix Tables 12-13]. **Sparse-Guard outperforms benchmarks by a factor of up to 6.6 under this attack.**
> - We also added 2 more state-of-the-art defense benchmarks including the new differential-privacy guaranteeing DP-SGD defense from Google DeepMind and Meta AI [Appendix Tables 10, 11, & 13].
>
> ***Across all (quadrupled) experiments, Sparse-Guard now outperforms benchmarks by a factor of up to 704. It also has the best PSNR (the most important metric) across every single experiment.***
>
> We absolutely agree with your conclusion that "an effective defense would be significant" on CIFAR-10 and similar hi-res data, and we believe we have addressed this concern.
>
> We also thank you for your comments that the paper is "well motivated and easy to follow" and your positive comments about our PyTorch codebase.

---

### Author Response · Authors · 2023-11-20
**Author Response to All Reviewers & Area Chair**

We thank all reviewers for their thoughtful and constructive reviews.

The main critique is that experiments should include more datasets (Reviewer CjY9, ETXH), higher-resolution datasets (Reviewer CjY9, ETXH), more defense benchmarks (Reviewer jF3v), and another type of attack (Reviewer jF3v, CjY9, ETXH).

**We have now addressed all reviewers reservations by *more-than-quadrupling* the number of experiments in the paper and we would appreciate it if you would improve your scores in light of our response.**
 Specifically, our expanded experiments show that **Sparse-Guard significantly outperforms all benchmarks (and in many cases, by a wider margin than in our original 2 datasets) on:**

- **~Two~ Three additional *datasets* (incl. *high-resolution data*) [Appendix Tables 8-13]:**

    - The ***high-resolution CIFAR-10 dataset*** [per Reviewer CjY9];
    - A dataset of ***medical images*** that represents a realistic 'worst-case' security application domain (Appendix Tables 8-13);
    - ***[Edit Nov. 21] CelebA celebrity faces*** (high-resolution) data [per Reviewer ETXH];

- **Two additional state-of-the-art benchmark *defenses* [Appendix tables 10, 11 & 13]:**
    - The ***Information Regularization defense of Wang et al.*** suggested by Reviewer ETXH;
    - The ***new differential-privacy guaranteeing DP-SGD defense of Hayes et al.*** that is currently under development at Google DeepMind and Meta AI, which is currently the only defense with provable guarantees for model inversion attacks;

- **The new *Plug-and-Play attack* of Struppek et al. suggested by Reviewers CjY9, jF3v, & ETXH** (Appendix Tables 12 & 13). This represents a 3rd evaluation setting (in addition to the end-to-end and split-network settings we consider). It is also the most recent high-profile attack paper in the literature.

**We run Sparse-Guard & *all* benchmarks on *all* 4 datasets (2 original + 2 new datasets), and ** then re-run Sparse-Guard & ***all 6***  **benchmarks (plus variants) under the new Plug-and-Play attack. EDIT Nov 21: We now added new results on the CelebA dataset (Tables 14-15). We emphasize that our new set of experiments on *[5 datasets, 6 benchmarks, 2 attacks, 3 settings]* is now significantly more comprehensive than other recent papers.** For example, the recent Plug and Play paper suggested by reviewers evaluates 3 datasets and 3 benchmarks.

***Across all (quadrupled) experiments, Sparse-Guard now outperforms benchmarks by a factor of up to 704. It also has the best PSNR (the most important metric) across *every single experiment*.*** In 2 of the 14 experiments, it loses to the Wang et al. defense in terms of SSIM by a negligible factor of 0.001, while significantly outperforming it in terms of PSNR and FID.

We also **thank reviewers for their comments that the paper is** *"novel" (Reviewers kXvF, jF3v, and ETXH), "well-motivated and easy to follow
 (Reviewer CjY9), "well-written and organized" (Reviewer kXvF), "well-motivated and well-written" (Reviewer ETXH), and "novel" (Reviewer kXvF)*, as well as all reviewers' *positive comments about our public PyTorch codebase*.

We emphasize Reviewer *ETXH*'s comment that our paper introduces a *novel sparse-coding research direction with many possible extensions to other important privacy settings*. We also emphasize Reviewer *CjY9*'s comment that a more effective defense against model inversion across all of the settings and data considered in our (now much larger) experiments constitutes a **significant contribution**. We also emphasize that there are immediate industry applications for a defense that is more practically effective---companies like Google DeepMind and Meta are actively working to integrate model inversion defenses into some of their AI products [Hayes et al., 2022].

**We respond to each reviewer separately below.**

---

> ### Comment · Reviewer_ETXH · 2023-11-20
>
> Dear authors,
>
> thanks for the rebuttal, additional experiments, and clarifications. Just a small remark: Plug and Play Attacks are not black-box but white-box since the attack exploits a model's gradients to optimize the GAN's latent vectors.

---

> ### Author Response · Authors · 2023-11-22
> **Extra Experiments on CelebA added Nov. 21**
>
> **We have re-uploaded our paper with even more experiments on the CelebA hi-res celebrity faces dataset suggested by Reviewer ETXH (this is our 5th dataset, and our 3rd added since the original submission).** Please see Appendix Tables 14 and 15. We note that this CelebA dataset is significantly larger in terms of resolution and image count, so compute times for all benchmarks are significantly greater (and the re-upload deadline is just hours away). Therefore, in the interest of time, we compare Sparse-Guard to the best of the benchmarks (Wang et al.) suggested by the reviewers, rather than all benchmarks. Sparse-Guard outperforms this best defense, and we also show that our performance advantage grows greater under the new Plug and Play attack (Table 15). We are adding remaining benchmarks as the finish computing, but we note that they tend to perform worse than the Wang et al. benchmark that we outperform here.
>
> **We strongly feel that a practically effective defense based on a completely novel research direction is of immediate interest both to other researchers and also to companies like Google and Facebook that are actively developing public products for sensitive application domains using available model inversion defenses.**

---

### Comment · Area_Chair_vYJu · 2023-11-20
**Comments on Authors' responses**

Dear Reviewers, The authors have responded to your valuable comments. Please take a look at their responses! Thanks!

---

### Meta-Review · Area_Chair_vYJu · 2023-12-05

**Metareview:**

The authors studied how a network's architecture contributes to its robustness (or vulnerability) by proposing
a novel sparse coding-based network architecture, Sparse-Guard, that is robust to model inversion attacks.

There are several rounds of active discussions between authors and reviewers, in particular with Reviewer ETXH. Still, there are several questions unsolved as the authors' responses did not satisfy Reviewer ETXH.
Although the authors commented that they have followed experimental settings from a main paper  (Titcombe et al. 2021), the additional request from Reviewer  jF3V is not unreasonable.
As commented by reviewers, it is recommended that this paper be revised (particularly the experimental section, which should put more focus on datasets like CelebA and a more detailed explanation of the experiment settings) and then resubmitted.

**Justification For Why Not Higher Score:**

The authors' responses did not satisfy most reviewers, in particular in experimental settings.

**Justification For Why Not Lower Score:**

none

---

### Decision · Program_Chairs · 2024-01-16

Reject